# The ratio of methanogens to methanotrophs and water-level dynamics drive methane transfer velocity in a temperate kettle-hole peat bog

Camilo Rey-Sanchez[1,4], Gil Bohrer[1], Julie Slater[2], Yueh-Fen Li[3], Roger Grau-Andrés[2], Yushan Hao[2], Virginia I. Rich[3], & G. Matt Davies[2]

[1]Department of Civil and Environmental Engineering and Geodetic Science, The Ohio State University, Columbus, Ohio, 43210, USA

[2]School of Environment and Natural Resources, The Ohio State University, Columbus, Ohio, 43210, USA

[3]Department of Microbiology, The Ohio State University, Columbus, Ohio, 43210, USA

[4]Current address, Department of Environmental Science, Management and Policy, University of California- Berkeley, California, 94720, USA

*Correspondence to*: Camilo Rey-Sanchez (rey.1@berkeley.edu)

**Abstract.** Peatlands are a large source of methane ($CH_4$) to the atmosphere, yet the uncertainty around the estimates of $CH_4$ flux from peatlands is large. To better understand the spatial heterogeneity in temperate peatland $CH_4$ emissions and their response to physical and biological drivers, we studied $CH_4$ dynamics throughout the growing seasons of 2017 and 2018 in Flatiron Lake Bog, a kettle-hole peat bog in Ohio. The site is composed of six different hydro-biological zones: an open water zone, four concentric vegetation zones surrounding the open water, and a restored zone connected to the main bog by a narrow channel. At each of these locations, we monitored water level (WL), $CH_4$ pore-water concentration at different peat depths, $CH_4$ fluxes from the ground and from representative plant species using chambers, and microbial community composition with focus here on known methanogens and methanotrophs. Integrated $CH_4$ emissions for the growing season were estimated as $315.4 \pm 166$ mg $CH_4$ m$^{-2}$ d$^{-1}$ in 2017, and $362.3 \pm 687$ mg $CH_4$ m$^{-2}$ d$^{-1}$ in 2018. Median $CH_4$ emission was highest in the open water, then decreased and became more variable through the concentric vegetation zones as the WL dropped, with extreme emission hotspots observed in the Tamarack mixed woodlands (Tamarack), and low emissions in the restored zone (18.8-30.3 mg $CH_4$ m$^{-2}$ d$^{-1}$). Generally, $CH_4$ flux from above-ground vegetation was negligible compared to ground flux (<0.4%), although blueberry plants were a small $CH_4$ sink. Pore-water $CH_4$ concentrations varied significantly among zones, with the highest values in the Tamarack zone, close to saturation, and the lowest values in the restored zone. While the $CH_4$ fluxes and pore-water concentrations were not correlated with methanogen relative abundance, the ratio of methanogens to methanotrophs in the upper portion of the peat was significantly correlated to $CH_4$ transfer velocity (the $CH_4$ flux divided by the difference in $CH_4$ pore-water concentration between the top of the peat profile and the concentration in equilibrium with the atmosphere). Since ebullition and plant-mediated transport were not important sources of $CH_4$ and the peat structure and porosity were similar across the different zones in the bog, we conclude that the differences in $CH_4$ transfer velocities, and thus the flux, are driven by the ratio of methanogens to methanotrophs relative abundance close to the surface. This study illustrates the

importance of the interactions between water level and microbial composition to better understand CH$_4$ fluxes from bogs, and wetlands in general.

## 1. Introduction

Methane (CH$_4$) fluxes from natural and anthropogenic sources play a significant role in determining atmospheric climate forcing (Ciais et al, 2013). Changes to CH$_4$ fluxes from natural systems are of significant concern due to their potential to drive positive feedback cycles in the global climate system (Bridgham et al., 2013; Dean et al., 2018). Natural wetlands emit approximately 30% of all the methane (CH$_4$) released to the atmosphere (Kirschke et al., 2013), yet, the uncertainty around wetland CH$_4$ flux is the highest of all the components of the global CH$_4$ budget (Kirschke et al., 2013). This uncertainty partly

arises from the complexity of physical and biological interactions that result in the production and oxidation of CH$_4$ and its eventual release to the atmosphere (Lai, 2009). Generally, water level (WL) is the most important driver of CH$_4$ emissions from wetlands, and especially peatlands, as its position in the soil or peat profile defines the boundary between anaerobic CH$_4$ production (methanogenesis) in the catotelm (i.e. the lower anoxic portion of the peat), and aerobic CH$_4$ oxidation (methanotrophy) in the acrotelm (the upper oxic peat) (Kettunen, 2003; White et al., 2008). However, a plethora of

environmental variables can also influence CH$_4$ fluxes in peatlands, including temperature (Bohn et al., 2007; Kim et al., 1999; Segers, 1998); peat origin (e.g. *Sphagnum*, woody peat, fen/reed peat) (Bridgham and Richardson, 1992); degree of humification (Glatzel et al., 2004); availability of labile carbon in the peat (Updegraff et al., 1995); concentrations of lignin, long-chain fatty acids, and polysaccharides along the peat profile (Hoyos-Santillan et al., 2016); phosphorous content, which regulates anaerobic decomposition of organic matter (Basiliko et al., 2007); the abundance of other electron acceptors, specially

Fe (Chamberlain et al., 2018); and pH, as methanogens occur at greater abundances in neutral to slightly alkaline conditions (Wang et al., 1993). It is also important to be cognizant of reports of CH$_4$ production in aerobic soil (Angle et al., 2017) and an increased awareness of the importance of anaerobic oxidation of CH$_4$ (Smemo and Yavitt, 2011).

The microbiota of a site can have complex interactions with WL and other physical conditions, which result in variable CH$_4$ fluxes. Despite the increasingly complex picture emerging of peatland CH$_4$ cycling, it has been estimated that methanotrophy

can oxidize 60-90% of the CH$_4$ produced in wetlands before it can escape to the atmosphere (Le Mer and Roger, 2001). Research has also shown that water table drawdowns reduce the abundance of methanogens (Kim et al., 2008), and that changes in ecosystem vegetation and structure can affect microbial community composition and, in turn, the CH$_4$ biochemistry of wetlands (McCalley et al., 2014). Generally, peat bogs are nutrient–poor sites dominated by hydrogenotrophic methanogenesis, but when disturbance occurs, a change from hydrogenotrophic to acetoclastic methanogenesis can occur due to an increase in

pH and nutrients (Kelly et al., 1992; Kim et al., 2008; Kotsyurbenko et al., 2004).

Kettle-hole peat bogs are peatlands created by the accumulation of peat in areas previously occupied by kettle lakes. Kettle-hole peat bogs, which are frequently found in Eastern North America (Cai and Yu, 2011; Moore, 2002), often consist of water

bodies surrounded by different vegetation zones. Closest to the open water there is often a mat of floating vegetation followed by concentrically organized vegetation zones that ultimately support shrubs and trees (Vitt and Slack, 1975). This vegetation heterogeneity can be an important driver of $CH_4$ fluxes (Lai et al., 2014), particularly in ombrotrophic peat bogs where vegetation communities and water levels are strongly associated (Malhotra et al., 2016). Measurements of $CH_4$ flux in different vegetation zones are important to understand site-level flux estimates at the bog scale that are affected by the relative cover and arrangement of different vegetation zones (Nadeau et al., 2013). Most importantly, a better understanding of the biological, chemical and physical processes controlling fluxes at these low resolutions are necessary to scale up $CH_4$ fluxes at the ecosystem level (Bridgham et al., 2013). The objectives of this study were to: 1) Calculate the growing-season $CH_4$ budget of a kettle-hole peat bog in Ohio by upscaling flux measurements from different vegetation zones. 2) Quantify the effects of biotic and abiotic controls on below-ground vertical profiles of $CH_4$ pore-water concentration and related fluxes. 3) Determine the links between microbial community structure and associated $CH_4$ dynamics. Brief comparisons of $CH_4$ dynamics between restored and undisturbed section are discussed but not in detail as the evaluation of the effect of restoration on $CH_4$ fluxes is not the objective of this paper.

## 2. Methods

### 2.1 Study site

We studied Flatiron Lake Bog, a ca. 14.4 ha kettle-hole peat bog located in north-eastern Ohio (41° 02' 40.67'' N 81° 21' 59.81'' W) (Fig. 1). The site is a State Nature Preserve and has been owned by The Nature Conservancy (TNC) since 1984. The greater part of the site typifies the characteristic abiotic and biotic zonation found in similar sites throughout Eastern North America. A small area (ca 1,120 $m^2$) of open water (Water) is located at the center of the site and is surrounded by a series of concentrically organized vegetation zones. The vegetation community of the site, including the bog vegetation and upland zones, was described in detail by Colwell (2009). The closest zone to the open water, hereafter called the Sphagnum-leatherleaf mat (Mat), consists of a floating mat of *Sphagnum fallax* (H. Klinggr.), with abundant cover of swamp loosestrife (*Decodon verticillatus* (L.) Elliot) and leatherleaf (*Chamaedaphne calyculata* (L.) Moench). Further away from the open water, and surrounding the Mat, is a narrow band of tamarack mixed woodland (Tamarack). The Tamarack zone is characterized by tamarack (*Larix laricina* (Du Roi) K. Koch) and yellow birch (*Betula alleghaniensis* Britton), with a ground layer dominated by *S. fallax*. Further towards the bog's periphery one finds a large area of mixed ericaceous shrubs (Shrubs) dominated by highbush blueberry (*Vaccinium corymbosum* L.), and huckleberry (*Gaylussacia baccata* (Wangenh.) K. Koch) with a ground layer of *Sphagnum* and scattered sedges, ferns forbs. The Shrubs zone also includes occasional patches dominated by winterberry (*Ilex glabra* (L.) A. Gray), or mature hardwoods, such as red maple (*Acer rubrum* L.) and yellow birch. Finally, the outermost area consists of a lagg or moat, hereafter called the winterberry lagg (Lagg). The Lagg is typically inundated during the first half of the growing season but dry during extended periods of the year. The dominant vegetation on the Lagg includes winterberry (*I. glabra*) and buttonbush (*Cephalanthus occidentalis* L.). The Water, Mat and Tamarack zones generally

present water levels that are always at (Water and Mat), or near (Tamarack) the surface, and together they are hereafter referred as the permanently-wetted area. In contrast, the Shrubs and Lagg zones have deeper water tables with more pronounced fluctuations in water level and are hereafter referred as the intermittently-wetted area. Peat coring and manual depth probing revealed a gradient in peat depths from the margin of the site to the interior. Measured peat depths varied from > 0.3 m in the

Lagg areas to > 10 m close the center of the site. The immediate upland area surrounding the bog is mostly forested with dominant tree species including American beech (*Fagus grandifolia* L.), black oak (*Quercus velutina* Lam.) and red maple (*Acer rubrum* L.). The width of this forested buffer varies, and some parts of the site are in proximity to areas under arable production, roads or buildings (Figure 1).

In addition to this relatively unaltered core area of the site, there is a restored section (Res) in the southern part of the bog,

which is connected to the main area by a narrow channel (Fig. 1) and comprises 19% of the total peatland (ca. 23,430 m$^2$). During the 1950s, this area was disturbed and drained to provide water for gravel and sand mining activities in adjacent areas. Peat coring in this area has revealed evidence of fire disturbance with significant deposits of charcoal and char layers. Between 2001 and 2003 TNC implemented a few restoration interventions in this area. This included opening of the channel to reconnect the two sections of the bog, and the installation of a water control structure to raise the water table at the restored section.

Elevated water tables suppressed red maple trees that had colonized the site since the disturbance and enabled the establishment of bog vegetation. The latter process was aided by the transfer of *Sphagnum* diaspores and the planting of *Vaccinium* spp. The current vegetation community for the restored section is dominated by winterberry (*I. glabr*a), buttonbush (*C. occidentalis*), invasive glossy buckthorn (*Frangula alnus* Mill.), and a remnant population of red maple trees. Thin discontinuous mats of *Sphagnum* spp., and *Carex* spp. dominated the ground layer. Due to its limited connection to the core of the site, and its history

of modification, degradation and restoration we consider the Res zone a distinct hydro-biological zone and due to its large variation in water level we consider this zone as part of the intermittently-wetted area as well.

## 2.2 Experimental design

Across the site, we established multiple sampling locations to assess ecosystem carbon fluxes, CH$_4$ pore-water concentrations, peat properties, water table dynamics and microbial community composition. Monitoring included locations within both the

undisturbed and restored sections of the bog. In the permanently-wetted area we initiated two transects with their start points located to the north and south of the open water in the center of the bog (Fig. 1). Each transect included three sampling locations, each associated with a vegetation zone: Water, Mat, and Tamarack. In the intermittently-wetted area, sampling locations for Shrubs and Lagg were selected as shown in Figure 1. Most locations were established in summer 2017 but the Tamarack location on the north transect and the Lagg location were added in the spring 2018. For the restored section two

randomly-selected locations were sampled, a northern location towards the center of the restored section (Res-N), and a southern location near the edge (Res-S). Fewer sampling locations in the restored section were justified by the more homogenous vegetation composition at the section scale, and the section's smaller area.

## 2.3. Surface CH$_4$ flux chamber measurements

In 2017, CH$_4$ gas transfer at the peat surface was measured monthly between June and October using non-steady state chambers. We sampled 2-4 chambers monthly in each sampling location at each zone. Chambers were deployed on top of semi-permanent collars that were installed 3 months prior to the first round of sampling. The collars in the Water, Tamarack, Shrubs and Res zones were made of rectangular high-density polyethylene (HDPE) boxes, with dimensions of 38 cm × 56 cm and a height of 26 cm. During sampling, the collars and the chambers had a foam-seal and were held together with clamps. For the open-water chambers, closed-cell polyethylene pipe insulation (1.3 cm internal diameter) was attached to the bottom edge of the chamber to facilitate flotation and create a seal with the water surface (Rey-Sanchez et al., 2018). For the Mat zone, we used tall chambers with a volume of 121 L (height 82 cm, radius 28 cm), with circular collars of 28 cm radius and a height of 59 cm that were inserted ca. 30 cm into the mat, for a total chamber height of ca. 121 cm. The height of the chambers was necessary to fit the tall and abundant loosestrife and leatherleaf plants. Due to their larger volume, these chambers included fans at 30 cm and 85 cm above the surface to improve air mixing within the chamber during sampling. The volume of the plants within the chamber was considered negligible.

All chambers included a thermometer to measure air temperature, a 3 m long Tygon tube (1.6 mm internal diameter) used as a vent for stabilizing pressure, and a 20 mm grey butyl stopper that served as a sampling port. In 2017 gas samples were extracted from the chambers using a syringe (30 ml). Here 20 ml of the gas sample were introduced into evacuated 10 ml vials to keep them over-pressurized. We used a closure time of 30 minutes for each chamber and extracted a sample every 5 minutes for a total of 7 samples per chamber. The gas extracted from the chamber was transported to the laboratory to be analyzed on a gas chromatograph (Shimadzu GC-2014, Shimadzu Scientific Instruments, Kyoto, Japan). Fluxes were calculated from the slope of the linear regression of the molar density of the greenhouse gas vs time. We incorporated selection criteria for rejecting outliers from individual chamber measurements as described in Morin et al (2017). Specifically, if the r$^2$ value of the linear regression of molar density vs time was not sufficiently high (r$^2$ ≥ 0.85), and the p-value was higher than 0.05, we removed one outlier point (identified as the point with the highest residual value) from the regression. This was done up to twice per chamber and if the accumulation rate regression still did not meet the selection requirements the entire chamber observation was rejected. This approach leads to the exclusion of cases where ebullition events occur during the sampling, creating a non-linear change in concentration. The procedure for calibration of the gas chromatograph is based on previous studies at the same facility (Nahlik and Mitsch, 2010; Sha et al., 2011) and was fully described in Morin et al (2017).

In 2018, surface fluxes were measured monthly and at the same locations as in 2017 and the additional Lagg and Tamarack locations. We used a portable infrared gas analyzer (Picarro GasScouter G4301, Picarro Inc, Santa Clara, CA) adapted to sample the same chambers as used in 2017. Given the higher sampling rate of the Picarro (1 point per second) the fluxes were calculated based on a linear regression of the molar density of CH$_4$ over 2-4 minutes depending on the volume of the chamber and the strength of the response of gas concentration vs time. Due to the higher number of points (146-293 per regression), a

stricter p-value was implemented ($p < 0.001$) to determine the significance of the regression. A lack of a significant correlation within a chamber measurements set was assumed to equal a zero flux.

Diurnal patterns of $CH_4$ emissions for the four main zones in the bog (O, Mat, Tamarack, and Shrubs) were measured in September in O-S, Mat-S, Tamarack-S, and Shrubs locations (Fig. 1). Four individual chamber measurements per location were completed throughout a full 24-hour cycle with a frequency of approximately 3 hours. Chamber measurements were accompanied by measurements of surface or water temperature, as appropriate.

### 2.4. CH4 flux from plants

To estimate potential emission of $CH_4$ through the plant tissues of larger sub-canopy and canopy trees and shrubs, which would be missed by chambers, we measured plant fluxes in dominant vascular species near the location of the surface measurements. Fluxes from plants were sampled monthly in June, July and September 2018 using the the Picarro gas scouter with chambers adapted to fit individual leaves or branch sections. Measurements were taken at multiple times during the day, in June, July and September, while a full diurnal pattern was performed in September.

To measure fluxes coming directly through the plant tissue in the Mat zone we used small chambers on loosestrife stems, the most abundant plant species in this zone. These chambers had a small opening in the corner of one the sections to allow the stem to sit uncompressed. The spaces around the stem hole were sealed with putty. This loosestrife-stem chamber enclosure had dimensions of 34 cm x 21 cm x 12.4 cm and a volume of 11.4 liters.

We used fully mature and healthy-looking loosestrife stems with more than 200 cm$^2$ of area for plant flux calculations. Stems were measured 5 times throughout the day in June and twice in July and September, adding up to 9 observations throughout the season. After 2-3 minutes of measurements, the stem was cut, wrapped in a moist paper towel, and put in a cooler for calculation of leaf area. The leaves were detached from the stem and petioles, arranged on a sheet of paper, and put on a scanner with a reference scale. The images were analyzed with the software ImageJ (Schneider et al., 2012) for calculation of total leaf area.

Plant-flux measurements at the Tamarack zone were conducted on stems and trunk sections of Tamarack, while fluxes at the Shrubs were measured from blueberry stems. To measure fluxes coming from trunk sections we used an adaptation of the chambers used by Pangala et al (2013) for tropical wetlands. These chambers had two sections sealed in between with insulation foam that closed around the trunk and that were held together tightly with clips. When holes around the trunk were present, additional layers of insulation foam were added to guarantee a good seal. The volume of the trunk inside the chamber was measured to subtract from the total volume of the chamber, which was 106 liters. The dimensions of all the enclosure were 76 cm x 112 cm x 52 cm. The understory fluxes from the low stems of the Tamarack as well as blueberry, the most abundant plant in the understory in the Tamarack and Shrubs zones, were measured using stem chambers. Trunk fluxes were measured six times in the months of May and July. For stem flux calculation, we used fully mature and healthy tamarack stems growing at a reachable height. Stems were measured twice in May, seven times in June, five times in July and twice in

September. Blueberry twigs were sampled at multiple locations within the Shrubs zones, four times in June, four times in July and twice in September.

## 2.5. Upscaling of CH$_4$ fluxes

To scale up the fluxes from each of the zones we extrapolated monthly mean chamber measurements to the entire area of each zone. We then integrated the monthly observed flux to calculate the total seasonal CH$_4$ budget for each zone and added the contribution of all the zones for the total seasonal site total. When fluxes from plants were significant, we calculated the total contributions by first multiplying the per-leaf-area rate observed by the plant chamber measurement by the leaf area index, then multiplying by the area of the zone, and finally integrating in time for the whole season. Leaf Area Index was calculated based on MODIS LAI product (Image Collection ID: MODIS/006/MCD15A3H, available through Google Earth Engine) for the period of study. Due to the low resolution of the imagery with respect to the site (500 m), we calculated the average LAI of the two images intersecting the site, which comprised similar areas.

Due to the lack of strength in the signal of the diurnal pattern, we did not correct the monthly measurements by time of day. The measurements in 2017 encompassed a total of 122 days for which the integration of fluxes was performed. The length of this period was higher in 2018 and was equal to 149 days.

## 2.6. Vertical profiles of CH$_4$ pore-water concentration and methane transfer velocity

We used *in-situ*, dialysis, pore-water samplers ("peepers") (Angle et al., 2017; MacDonald et al., 2013) to measure vertical pore water concentration profiles of dissolved CH$_4$. In total, seven peepers were installed throughout the site: five in the undisturbed section and two in the restored section. Peepers were placed adjacent to the gas flux chambers. Each peeper had 10 sampling windows located at depths from 1.4 to 51.8 cm and spaced every 5.6 cm. Each window (8.89 $\times$ 2.28 cm$^2$ area, 3.02 cm depth), which was filled with DI water that equilibrates with the surrounding pore water through a semi-permeable membrane (pore size 0.2 µm) (Sterlitech Corporation, Kent, WA), was connected to two UV-resistant tygon tubes that extended to the surface. From one tube water was suctioned using a syringe, the other was connected to a nitrogen bag to replace the volume of water extracted. Extracted samples were stored in 10 ml glass vials, each containing 100 µl of hydrochloric acid (2M) to prevent any biological reactions. Samples were kept in a cooler at low temperatures (ca. 4° C) for no longer than two days before processing.

Samples were processed with the goal of measuring the concentration of dissolved gases in the water. 5 ml of water sample were extracted from each vial and placed in a syringe pre-filled with 20 ml of N$_2$ gas. The syringes were shaken vigorously for 15 min and 20 ml of the headspace was extracted into a new 10 ml glass vial. The pore-water concentrations of the samples were calculated based on the headspace concentration of the gas in equilibrium with the liquid sample according to Henry's law of equilibrium of gases in a liquid-air interface. The coefficient of equilibrium for CH$_4$ was 67.13 L MPa mol$^{-1}$. The gas samples were analyzed in a gas chromatograph with a FID detector (Shimadzu GC-2014, Shimadzu Scientific Instruments, Kyoto, Japan).

By combining pore-water concentration at the surface with the associated fluxes, estimations of methane transfer velocity were obtained as in previous studies in forested ponds and lakes (Holgerson et al., 2017; Schilder et al., 2016; Wanninkhof, 2014). Through this approach, the flux at the water-air interface can be calculated using the bulk formulation:

$$FCH_4 = k(C_w - C_{eq})$$ Eq. (1)

Where $FCH_4$ is the diffusive $CH_4$ flux (mol m$^{-2}$ s$^{-1}$), $k$ is the $CH_4$ transfer velocity (m s$^{-1}$), $C_w$ is the concentration of methane in the porewater at the surface (mol m$^{-3}$), and $C_{eq}$ is the concentration of $CH_4$ in equilibrium with the atmosphere (mol m$^{-3}$). $C_{eq}$ is calculated by multiplying the mixing ratio of $CH_4$ in the atmosphere ($r$, in mol mol$^{-1}$) by the atmospheric pressure ($P$, in MPa) and dividing by Henry's Law coefficient of equilibrium for $CH_4$ ($K_H$) of 0.067 m$^3$ MPa mol$^{-1}$ as in eq. 2:

$$C_{eq} = \frac{r\,P}{K_H}$$ Eq. (2)

$C_{eq}$ was calculated first with a constant $r$ (2 μmol mol$^{-1}$) and second with the value of the average of the initial $r$ of the chamber measurements associated with each flux calculation. These two methods produced nearly identical results in $C_{eq}$ when compared to the much higher values of $C_w$. The constant mixing ratio was chosen for the rest of the analyses given the uncertainty associated with the initial concentration from the chambers. In the case of our peat bog, $C_w$ can be calculated by multiplying pore-water concentration ([$CH_4$]) by peat porosity, $\Phi$ (see ancillary measurements below):

$$C_w = [CH_4]\Phi$$ Eq. (3)

Where [$CH_4$] was calculated in the top stratigraphic layer of the peat (ca. 10 cm). Finally, methane transfer velocity can be calculated as:

$$k = \frac{FCH_4}{C_w - C_{eq}}$$ Eq. (4)

We focus on the top 10 cm because, first, this is the section where the atmospheric exchange occurs. Secondly, this section should be the most active one for both methanogens and methanotrophs (Angle, 2017) since it includes the more aerobic acrotelm as well as less well-humified peat (greater labile C availability).

**2.7. Core sampling, DNA extraction, 16S rRNA amplicon sequencing and analysis**

We analyzed the microbial composition of peat cores adjacent to the peepers. Three cores were extracted in August 2017 from within 5 meters of the peepers located in the Mat-S, Tamarack-S, Shrubs, Res-N and Res-S zones. The cores were extracted using a rectangular Wardenaar peat corer with an aperture area of 12 × 12 cm and >50cm length. Core horizons were sampled in the field according to obvious stratigraphy (by color, texture and Von Post humification). Representative ca. 10 cm long samples of each horizon were stored at 4 °C and processed the next day for microbial analyses. Processing involved dividing

each section vertically into three sub-samples, which were homogenized before a 0.25g sub-sample was extracted from each. A fourth 0.25 g sub-sample was taken following homogenization of all the remaining material from a given section. All sub-samples were stored at -20 °C for no more than three months until DNA extraction. DNA was extracted using DNeasy PowerSoil kit (Qiagen, Hilden, Germany) following the manufacturer's protocol. Extracted DNA was quantified with Nanodrop 8000 (Thermo Fisher Scientific, Waltham, WA). The 16S rRNA V4 region was then amplified and sequenced on the Illumina MiSeq platform (Illumina, San Diego, CA), at Argonne National Labs, via the Earth Microbiome Project (http://www.earthmicrobiome.org/) post-2015 barcoded primer set. These primers (515F (Parada) CGTGYCAGCMGCCGCGGTAA – 806R (April) GGACTACNVGGGTWTCTAAT, forward-barcoded; Parada et al., (2016) and Apprill et al., (2015)) are adapted for Illumina HiSeq2000 and MiSeq by the addition to the forward primer of a 5' Illumina adapter to support paired-end sequencing, a twelve-base barcode sequence to support sample pooling in each lane, and forward pad and linker sequences, and the addition to the reverse primer of a 3' Illumina adapter, and reverse pad and linker sequences (Caporaso et al, (2010), redesigned by Walters et al. (2016)). Each 25µl PCR reaction contained 12µl of MoBio PCR water (Certified DNA-Free), 10µl of 5Prime HotMasterMix (1x), 1µl of forward primer (5µM concentration, 200pM final), 1µl Golay barcode-tagged reverse primer (5µM concentration, 200pM final), and 1µl of template DNA. The conditions for PCR were: 94°C for 3 minutes to denature the DNA, with 35 cycles at 94 °C for 45 s, 50 °C for 60 s, and 72 °C for 90 s; with a final extension at 72 °C for 10 min to ensure complete amplification. The PCR amplicons were quantified using PicoGreen (Invitrogen, Carlsbad, CA) and a plate reader. Once quantified, various volumes of each of the amplicons were pooled into a single tube for equal representation of each sample. This pool was then cleaned using UltraClean PCR Clean-Up Kit (MO BIO Laboratories, Inc.), and quantified using the Qubit (Invitrogen, Carlsbad, CA). After quantification, the molarity of the pool was determined and diluted to 2nM, denatured, and then diluted to a final concentration of 4.0pM with a 10% PhiX spike for sequencing on the Illumina MiSeq, via the 2x150bp protocol.

Sequence data were processed with the bioinformatic software QIIME 1.9.1 (Caporaso et al., 2010) using 16S-RDS pipeline (Nelson et al., 2014) with slight modifications. The subset of amplicon-based lineages identified as genera of known methanogens and methanotrophs (Appendix A, Table A1) were then further profiled for this study. Sub-samples were averaged to obtain one mean value for each section within each core.

**2.8. Ancillary measurements**

Data from near-by NOAA meteorological stations WBAN:14813 and WBAN:14985 (https://www.ncdc.noaa.gov/cdr) were used to obtain hourly and daily averages of air temperature, precipitation and atmospheric pressure. Eight dip-wells adjacent to the peepers (Mat-N, Mat-S, Tamarack-N, Tamarack-S, Shrubs, Lagg, Res-N, Res-S) were used for monthly measurements of water level. Water level was measured continuously between June 2017 and October 2018 in four of the eight dip-wells (Mat-S, Tamarack-S, Shrubs, Res-S). Water levels at other locations were estimated based on an offset between manual readings of water level. To calculate water levels we used HOBO pressure sensors (Onset computer corporation, Bourne, MA) that were corrected using atmospheric pressure data from the NOAA stations. Adjacent to each peeper, we measured vertical

profiles of dissolved oxygen 2-4 times a year using a probe equipped with a fiber optic sensor and a temperature sensor (PreSens Precision Sensing GmbH, Regensburg, Germany). The probe was inserted to a depth of 80 cm and allowed to stabilize for ca. 30 min. The probe was then moved upwards in 10-20 cm increments to complete a profile up to the level of the water table.

Water from the eight dip-wells was sampled for chemical analysis roughly monthly between 27 June and 3 October 2017 in four sampling events. All dip-wells were perforated at 60-90 cm below ground level to ensure that water samples were collected from a consistent depth. Before collecting water samples, dip-wells were emptied completely using a vacuum syringe and allowed to refill. When it was impossible to completely empty a dip-well due to rapid recharge, a volume of water equivalent to the volume of the dip-well was removed before collecting samples. EC and pH were measured in the field using a YSI

Pro1030 pH, conductivity and salinity instrument. When dip-well recharge was insufficient for EC and pH measurements in the field, these measurements were made within 48 hours in the lab using a YSI EcoSense EC30A conductivity and TDS pen tester and a YSI EcoSense EH10A pH/temperature pen tester. Water samples were then filtered using Whatman binder-free glass microfiber 0.7µm filters that had been combusted at 500ºC to remove organic contamination. Water samples were stored in HDPE coated bottles and frozen at -22ºC for 10 months prior to analysis. Inductively coupled plasma -optical emission

spectrometry (ICP-OES) (US EPA, 2015b) was carried out using a Varian Vista-MPX to measure concentration of Al, Ca, Fe, K, Mg, Mn, Na, P, S, and Zn. Concentrations of $NO_3+NO_2$ nitrogen (measured as a combined value) and $NH_4$ nitrogen were determined by colorimetry using Lachat's QuikChem® 8500 Series 2 Flow Injection Analysis System (US EPA, 2015a). Quality assurance and quality control protocols were followed for both the ICP-OES and flow injection anlayses. Recoveries of matrix spikes and serial dilutions were at least 75% and 90%, respectively. The reporting limit (RL) for each batch of

samples was the lowest concentration in the calibration curve. The RL for $NH_4$-N was 0.1 mg/L and the RL for all other analytes was 0.01-0.05 mg/L. Where concentrations were below the reporting limit, the measured concentration was substituted with one-half the reporting limit. Check standards and blanks were analyzed every 10 samples. Check standard recoveries did not exceed +/-10% error and blanks did not exceed reporting limits. No blanks were allowed to exceed the reporting limits. Accuracy of pH and EC measurements was ensured through regular calibration of equipment.

In June 27, 2017, one core from the Shrubs zone and one from the Res zone were extracted for analysis of peat bulk density and porosity. The core was sliced every 2.5 cm to a depth of 50 cm. Samples were packed and sealed in plastic bags and taken to the laboratory to measure wet weight. Samples were then dried in an oven at 60 ° C for 2-3 days until the weight was stable. Peat bulk density was calculated based on the weight of dry soil occupied by slices of 2.5 x 12 x 12 $cm^3$. Porosity was calculated as 1 minus the ratio of peat bulk density to soil particle density, which was estimated as 1.45 Mg $m^{-3}$ for *Sphagnum* peat soils

(Oleszczuk and Truba, 2013).

**2.9. Data analysis**

Data preparation was completed in Matlab (R2017b, Mathworks), and statistical analyses in R version 3.5.1 (R Development Core Team, 2018). Differences in $CH_4$ fluxes between hydro-biological zones were evaluated using a linear mixed-effects

model (lmm) through the function "lmer" implemented in R in the package "lmerTest" version 3.0-1 (Kuznetsova et al., 2017). Transformation of $CH_4$ flux data to their logarithm base 10 was applied to improve the normality of the data and the normality of the residuals of the model. The fixed effects in the model were *Zone* (Water, Mat, Tamarack, Shrubs, and Res), a categorical value for year (*Year*), a categorical value for the month of measurements (*Month*), temperature 10 cm below the surface (*Tsurf*), mean water level for a month before the flux measurements (*WLm*), and a continuous variable representing the time to noon in hours (*t2noon*). *Transect* (North or South) was specified as a random effect. We also tested for the interactions between *Zone* and *Month*, *Zone* and *WLm,* and *Zone* and *Tsurf* but they were not significant. The final statistical model for both $CH_4$ flux is described in Eq. (5):

$$Flux \sim \ Zone + \ Tsurf + WLm + Year + Month + t2noon + (1|Transect) \qquad \text{Eq. (5)}$$

Pair-wise differences in emissions among the zones were evaluated through testing differences in the marginal means of the reference grid of the mixed model using the package "emmeans" in R (Lenth et al., 2018). The overall effect of the factors within the model was evaluated with an ANOVA of the model. Significance in the model was defined with a p value of 0.05. To evaluate if plant fluxes were significantly different from zero we used a one-sample Wilcoxon test.

Pore-water concentrations of $CH_4$ were evaluated using a linear mixed model. We used a similar model to evaluate pore-water CH4 concentrations, except that we added depth to the surface (Depth) as a fixed effect (see Eq. 6). We deleted the interaction between depth and zone because it was not significant. The final model for pore-water concentrations of $CH_4$ is described in Eq. (6):

$$CH_4 \ Pore \ water \ conc. \sim Zone + Depth + WLm + Tsurf + Year + Month + (1|Transect) \qquad \text{Eq. (6)}$$

Pair-wise differences in pore-water concentrations between zones were tested by evaluating differences in the marginal means in the same way as for the model of $CH_4$ flux. The overall effect of the factors in the model was evaluated with an ANOVA. Chemical analyses were not included in the model as chemistry data was only available for 2017. Instead, a principal component analyses was run on the chemical variables (12 chemical species plus EC and pH, (Appendix A, Table A2) at the 8 sampling locations, and the scores of the first principal component were correlated to mean $CH_4$ fluxes, mean $CH_4$ pore-water concentration, and mean $CH_4$ transfer velocity. Differences in element concentrations between different vegetation zones and between different locations were evaluated using ANOVA. Pair-wise comparisons were evaluated using a Tukey-HSD Post-hoc test. Differences in peat bulk density were evaluated using an ANCOVA of zone and depth. The relationships between microbiota and methane fluxes was evaluated through a correlation of the ratio of the relative abundance of methanogens to the relative abundance of methanotrophs versus, $CH_4$ flux, mean $CH_4$ pore-water concentration, and $CH_4$ transfer velocity.

# 3. Results

## 3.1. Inter- and intra-annual variation in abiotic conditions

The mean air temperature during the growing season (May 1st to Oct 31st) was 20.4 °C in 2017 and 22.5 °C in 2018 as measured by standard meteorological stations. In 2017 and 2018, total precipitation for the growing season was 196 mm and 356 mm, respectively (Fig. 2). Water level ranged from -45.4–19.7 cm in 2017 and -55.1–27.3 cm in 2018, where negative levels indicate a water table below the ground surface (Fig. 2). As expected, the intermittently-wetted area (Shrubs + Lagg + Res zones) experienced substantial fluctuations in water level, while in the permanently-wetted area (Tamarack) the water level remained at or close to the surface (Fig. 2). Fluctuations were smaller in the Tamarack zone, with the water table drawing-down to a maximum depth of 12 cm compared to a maximum of 53 cm in the Shrubs (Figure 2).

pH was similar throughout the bog, with higher values occurring in the restored (Res) zone than in the undisturbed zone (Appendix A, Table A2), but with no significant differences among the hydro-biological zones (F = 0.98, p = 0.43). The Lagg zone had significantly higher concentrations of Fe, Ca, Mg, and Mn when compared to other hydro-biological zones (p < 0.05 for all paired relationships). The restored section had significantly higher concentration of Mn (F = 3.80, p = 0.01) and Na (F = 3.78, p = 0.01). Concentrations of Ca and P tended to be higher in the restored section as well, however, the differences were not significant when comparing all hydro-biological zones (F = 2.88, 2.47; p = 0.05, 0.07; respectively). Interestingly, concentrations of ammonia ($NH_4^+$) were significantly higher in the Tamarack zone (F = 10.6, p < 0.001) than in all the other zones, while concentrations of nitrate (NO3-) were generally low and did not significantly differ among zones (F = 0.05, p = 0.91).

The northern section of the bog collected runoff from adjacent agricultural fields, and consequentially, had higher pH, electrical conductivity and concentration of elements including S, Na, Mn, Mg, Fe and Ca than the rest of the bog. Notably, when considering location-wise comparisons, the concentrations of S, Ca, and Mn were significantly higher in the Tamarack-N than in all other locations of the undisturbed bog (Appendix A, Table A2). Location differences also occurred in the restored section. pH was significantly higher in the Res-N location (p < 0.05) and P was significantly higher in the Res-S location (p < 0.05).

Vertical profiles of dissolved oxygen confirmed the existence of anoxic conditions below the water level. Dissolved oxygen concentrations below the water level were always less than 0.1 mg l$^{-1}$, whereas above the water level the concentration increased sharply. The only exception were the profiles taken at the Mat, which had an average dissolved oxygen concentration of 0.27 mg l$^{-1}$.

Peat bulk density was significantly lower in the Shrubs than in the Res zone (F = 34.5, p < 0.001), with averages ± SD of 0.08 ± 0.02, and 0.12 ± 0.03 g m$^{-3}$, respectively. Calculated porosities assuming a peat particle density of 1.45 g m$^{-3}$ (Oleszczuk and Truba, 2013), were equal to 94.5 and 91.8%, respectively. Because the peat was saturated at the time of extraction this porosity is equivalent to the volumetric water content. There was not a significant effect of depth on peat bulk density (F = 0.05, p = 0.82).

## 3.2. The effect of different hydro-biological zones and water level on $CH_4$ emissions

There were higher $CH_4$ emissions towards the central, permanently-wetted part of the bog (Table 1). The fluxes from the Lagg zone were not significantly different than zero (t-test, p = 0.185) and were therefore excluded for future comparisons among zones. Mean $CH_4$ fluxes were significantly different between hydro-biological zones (F = 1.14, p <0.001). The fluxes from the Water zone were not significantly higher than the fluxes from the other units within the permanently-wetted area (t-ratio = - 1.45 p= 0.59; and t-ratio = 1.27, p = 0.70; for Mat and Tamarack, respectively), but they were significantly higher than fluxes in the intermittently-wetted area (Shrubs zone: t-ratio = 5.83, p < 0.001, Res zone: t-ratio = 6.53, p < 0.001). $CH_4$ emissions from the restored section were significantly lower than the emissions from units in the permanently-wetted area, the Mat (t-ratio = -4.6, p < 0.001), and the Tamarack zones (t-ratio = -6.1, p < 0.001). However, $CH_4$ emissions from the restored section were not significantly different to the emissions from the Shrubs zone (t-ratio = -0.17, p= 0.99).

Mean water level (*WLm*) had a significant effect on $CH_4$ flux (F = 8.49, p= 0.003), with higher emissions occurring when *WLm* was more positive (higher *WLm*). The effect of water level on $CH_4$ fluxes was not significant when considering instantaneous water levels at the time of the measurements but was significant when considering the average water level data throughout 30 days prior to the flux measurement. The effect of temperature (*Tsurf*) was not significant (F = 0.71, p= 0.40).

## 3.3. Temporal variations in $CH_4$ fluxes

There was a substantial temporal variability in $CH_4$ fluxes. The open water zone was the only zone that had a distinct and consistent seasonal cycle, where the fluxes increased from May to the middle of the growing season, peaking in early September and declining in October (Figure 3). In the Tamarack zone, fluxes declined over the growing season in 2017, but in 2018 the flux peaked in early September, where there were two extremely high flux measurements at the northern transect of 27,180 and 8,605 nmol $m^{-2}$ $s^{-1}$ that skewed the average to a total of 6,748 nmol $m^{-2}$ $s^{-1}$. There was no significant relationship between month of measurement and $CH_4$ flux (F = 2.21, p = 0.05). Across all hydro-biological zones, $CH_4$ fluxes were not significantly different in 2017 and 2018 (F = 2.59, p = 0.11).

Although the relationship of $CH_4$ emissions with time to noon was significant (F = 13.1, p < 0.001), the diurnal measurements from September 2018 (Fig. 4) did not indicate strong diurnal patterns of $CH_4$ emissions. In the open water zone, $CH_4$ emissions decreased during the late afternoon-early evening, which approximately coincided with a peak in water surface temperature (Fig. 4). In the Tamarack zone emissions increased with warmer temperatures in the afternoon. In the Mat zone there was a peak in the middle of the morning but there was no apparent relationship with surface temperature. There was no clear diurnal pattern of $CH_4$ emissions in the Shrubs zone, likely a consequence of very low $CH_4$ emissions during the time of measurements.

## 3.4. Plants fluxes and upscaling of $CH_4$ emissions

Fluxes from plant tissues were negligible compared to the fluxes from the peat or open water surfaces (Fig. 5). Measurements from Loosestrife, the most abundant vascular plant in the Mat, and from tamarack stems and stems were not significantly

different from zero (p = 0.83; p = 0.48; p = 0.06, respectively). Fluxes from the blueberry leaves were significantly different than zero (p = 0.01) and averaged -1.11 nmol m$^{-2}$ s$^{-1}$, indicating net uptake of $CH_4$ by or through blueberry plants (Fig. 5).

The peat bog emitted a total of 4.8 ± 1.9 and 5.5 ± 8.4 Tons of $CH_4$ during the growing seasons of 2017 and 2018, respectively. The high uncertainty in 2018 was due to the larger variation of fluxes produced by high fluxes in the Tamarack zone, which emitted a total of 0.12 ± 0.14 Tons of $CH_4$ in 2017, but a much higher 5.4 ± 8.2 Tons of $CH_4$ in 2018.

Blueberry leaves acted as a slight sink of atmospheric $CH_4$ with a mean flux of -1.11 nmol m$^{-2}$ s$^{-1}$. The total sink of $CH_4$ from blueberry bushes was equal to -46.9 ± 20 and -57.4 ± 24 Kg of $CH_4$ for 2017 and 2018, respectively. These values were equal to a small offset of the total daily emissions, by 0.37% for 2017, and 0.14% for 2018.

Because the length of the measurement periods in the growing seasons were not equal among years, total emissions (Table 1) were divided by the length of the measurements period to produce estimates of mean total flux per day. These values were then divided by the area of the bog (excluding the Lagg zone) to produce the final result of 315.4 ± 166 mg $CH_4$ m$^{-2}$ d$^{-1}$ in 2017 and 362.3 ± 687 mg $CH_4$ m$^{-2}$ d$^{-1}$ in 2018.

### 3.5. Dissolved $CH_4$ pore-water concentrations and methane transfer velocity

Excluding the Mat zone, the mean $CH_4$ pore-water concentration per zone followed a pattern similar to the fluxes, with higher concentrations in the Tamarack zone, followed by Shrubs, Res, and Lagg zones (Fig. 6). Pore-water $CH_4$ concentrations were significantly higher in the Tamarack zone than in the Mat zone (*t*-ratio = 3.3, p = 0.003) and in the Shrubs zone (t-ratio = 6.4, *p* < 0.001). Pore-water $CH_4$ concentrations were significantly lower in the Res zone than in the Mat zone (t-ratio = -7.2, p < 0.001), the Tamarack zone (t-ratio = -17.1, p < 0.001), and the Shrubs zone (t-ratio = -6.8, p < 0.001), but not significantly different from concentrations in the Lagg (t-ratio = 0.28, p = 0.77; Fig. 6). Differences in $CH_4$ pore-water concentration between Mat and Shrubs zones were not significant (t-ratio = 1.98, p = 0.19). It is important to note that times for which the water table was below the level of a certain peeper sampling window, and thus there was no pore water at that given height, were considered as missing values.

There was a significant relationship between $CH_4$ concentration and depth (F = 85.3, *p* < 0.001), with pore-water concentrations of $CH_4$ increasing with depth. $CH_4$ pore-water increased significantly with increasing temperature 10 cm below the surface at the time of measurement (*Tsurf*) (F = 20.9, p < 0.001), and with the average water level during the month preceding the measurement (*WLm*) (F = 16.2, p < 0.001). Higher water tables were associated with increased $CH_4$ pore-water concentration throughout the whole profile.

Per location, average (mean ± SD) $CH_4$ pore-water concentration in the top 50 cm of the peat was the highest in Tamarack-S (0.86 ± 0.62 mM), followed by Tamarack-N (0.76 ± 0.36 mM), Shrubs (0.30 ± 0.26 mM), Mat-N (0.21 ± 0.12 mM), Mat-S (0.19 ± 0.12 mM), Res-N (0.14 ± 0.13 mM), Lagg (0.10 ± 0.08 mM), and Res-S (0.09 ± 0.08mM). Ammonium concentration was positively correlated with $CH_4$ pore-water concentration averaged for the whole profile ($r^2$ = 0.70, p = 0.005) and for the top peat layer ($r^2$ = 0.83, p < 0.01).

CH$_4$ pore-water concentrations were significantly different among months, with concentrations always lower in May (p < 0.001 for all paired relationships), and June (p < 0.001 for all paired relationships), and higher in August, around the peak of the growing season, and October, at the end of the growing season. CH$_4$ pore-water concentrations were significantly higher in 2018 than in 2017 (F = 24.9, p < 0.001).

Overall, there was no significant relationship between average concentration and surface fluxes (r$^2$< 0.01, p = 0.95), even when considering only the top layers of the peat column, where a better relationship was expected (r$^2$ = 0.08, p = 0.11) (Fig. 7a). The lack of a relationship was surprising as the values included only those times at which the top stratigraphic layer of the peat was saturated. Methane transfer velocity was calculated from these data (Fig. 7b) and from the times when the microbiology data was available to compare against (Fig. 7c).

The first principal component (PC1) of the chemical analytes explained 37.6% of the variation in the dataset while the second explained 28.5% (Appendix B, Fig. B1). The ten variables that contributed the most to PC1 were, in order, Mn, Ca, Mg, S, P, Al, ec, NA, NO3, and K. There was no significant relationship between PC1 and CH$_4$ flux (r$^2$ = 0.17, p = 0.17), CH$_4$ pore-water concentration (r$^2$ = 0.15, p = 0.80) or CH$_4$ transfer velocity (r$^2$ = 0.12, p = 0.65).

**3.6. Both methanogens and methanotrophs were more abundant in permanently-wetted zones**

Overall, both methanogens and methanotrophs were at higher relative abundances (as a portion of the overall microbial communities) in the permanently-wetted zones Mat-S and Tamarack-S (where they accounted for 1.8 and 2.0 % of the microbial communities respectively, by amplicon percentages), than in the intermittently-wetted zones Shrubs, Res-N, and Res-S (0.2, 0.1 and, 0.1 %, respectively) (Fig. 8). In addition, hydrogenotrophic methanogens (*Methanobacterium* and *Methanoregula*) were much more abundant than acetoclastic methanogens (*Methanosaeta* and *Methanosarcina*) at all sites

(Fig. 8). Among the hydrogenotrophs, *Methanobacterium* was broadly present, while *Methanoregula* was generally a larger component of the methanogen community in saturated, undisturbed peat (Mat-S, Tamarack-S, and deep Shrubs). Among the acetoclasts, *Methanosaeta* was observed only in the permanently-wetted zone Mat-S and Tamarack-S, and accounted for a small proportion of total methanogens except at 50cm in Mat-S. In the restored zones, where acetoclasts had higher relative abundances, the genus *Methanosarcina* was predominant.

Methanotrophs were mostly present in the permanently-flooded zones Mat-S and Tamarack-S and were particularly abundant in peat strata closer to the surface (0-20 cm). *Methylomonas* accounted for most of the methanotroph sequences found in this study, and dominated the methanotrophs of the Mat-S and Tamarack-S zones, while *Methylosinus* was a much larger portion of the methanotrophs in the Shrubs and Res zones (Fig. 8) even as overall methanotroph relative abundance dropped to less than 0.05% of the microbial community.

Methane fluxes were not correlated to the relative abundance of methanogens (r$^2$ = 0.01, p = 0.74) or methanotrophs (r$^2$ = 0.01, p = 0.78). In addition, mean CH$_4$ concentrations were also not correlated to the relative abundance of methanogens (r$^2$ = 0.01, p = 0.83) or methanotrophs (r$^2$ = 0.01, p = 0.70). However, for the principal coordinates analysis of sites based on geochemistry, PC1 was significantly negatively correlated to methanogens' relative abundance (r$^2$ = 0.90, p < 0.01). As

indicated above, most of the variation in PC1 was driven by Mn, Ca, Mg and S, and there was a significant relationship between mean methanogen relative abundance and manganese ($r^2 = 0.90$, p = 0.007) and sulfur concentrations ($r^2 = 0.74$, p = 0.03). When considering only the bottom 25 cm of the peat profile, the layer from which pore water was taken for chemical analyses, methanogen relative abundance was negatively correlated to electrical conductivity ($r^2 = 0.85$, p = 0.01). In the top layer of the

peat, where methanotrophs are more active, there was a negative correlation between methanotroph relative abundance and magnesium concentration ($r^2 = 0.79$, p = 0.03).

## 4. Discussion

### 4.1. The $CH_4$ budget and its heterogeneity among hydro-biological zones

There were relatively high $CH_4$ emissions in Flatiron Lake Bog compared to previously reported fluxes in other northern

peatlands. Average daily $CH_4$ emissions were equal to $315.4 \pm 166$ mg $CH_4$ $m^{-2}$ $d^{-1}$ in 2017 and $362.3 \pm 687$ mg $CH_4$ $m^{-2}$ $d^{-1}$ in 2018. These values were higher than emissions in ombrotrophic peat bogs in Minnesota (monthly average range: 27-240 mg $CH_4$ $m^{-2}$ $d^{-1}$) (Chasar et al., 2000), (117 mg $CH_4$ $m^{-2}$ $d^{-1}$)(Dise, 1993), and Michigan (0.6-209 mg $CH_4$ $m^{-2}$ $d^{-1}$) (Shannon and White, 1994), and a boreal bog in Northern Quebec (57 mg $CH_4$ $m^{-2}$ $d^{-1}$) (Nadeau et al., 2013). Higher $CH_4$ fluxes compared to other bogs is likely the result of the higher temperatures experienced in Ohio, which is at the southern limit of Northern

peatland distribution.

Methane fluxes were highly heterogeneous, with a variation of over 4 orders of magnitude and with a skewed distribution due to extreme events of $CH_4$ flux (median: 33.7 nmol $m^{-2}$ $s^{-1}$, range: -12.2 – 27186 nmol $m^{-2}$ $s^{-1}$). The skewed distribution of $CH_4$ fluxes and heterogeneity has also been found by Christen et al. (2016) in a Canadian undisturbed scrub-pine *Sphagnum* bog (median 42 nmol $m^{-2}$ $s^{-1}$, range 5–3500 nmol $m^{-2}$ $s^{-1}$), and by Treat et al. (2007) in a temperate fen in New Hampshire (range:

6.3–2772 nmol $m^{-2}$ $s^{-1}$). We found higher emissions in the open water (mean 122, median 61.9, range: 0.14–1823 nmol $m^{-2}$ $s^{-1}$) than in the other hydro-biological zones. This pattern was also found by Christen et al. (2016), who found that fluxes from open waters or ponds had an average of 3336 nmol $m^{-2}$ $s^{-1}$ and a median value of 2670 nmol $m^{-2}$ $s^{-1}$ compared to collars on the ground containing vegetation that had mean and median values of 986 and 47 nmol $m^{-2}$ $s^{-1}$, respectively. On an analysis of a variety of peatlands in Minnesota Crill et al. (1988) also found that mean $CH_4$ emissions were 294 mg $m^{-2}$ $d^{-1}$ in open bogs,

while in forested bogs the mean was equal to 77 mg $m^{-2}$ $d^{-1}$. This result agrees with our calculations, where we find daily normalized fluxes averaged for both years of 279 mg $CH_4$ $m^{-2}$ $d^{-1}$ in open water and 224.72 mg $CH_4$ $m^{-2}$ $d^{-1}$ in the mixed ericaceous shrub units.

There were extremely-high $CH_4$ flux measurements from the northern transect of the Tamarack zone in September 2018 (27180 and 8605 nmol $m^{-2}$ $s^{-1}$) and in October 2018 (2808 and 6609 nmol $m^{-2}$ $s^{-1}$). These measurements were not ebullition events

since the increase in concentration with time was steady (Appendix B, Fig. B2) and the coefficient of correlation for both flux events was higher than 0.97. They were not localized events either since the two collars were about 1.5 m apart from each other. Unfortunately, a core was not taken at the northern transect were this event occurred so the abundance of methanogens

and methanotrophs could not be tested. Interestingly, the concentration of sulfur was significantly higher in this zone indicating that the Tamarack-N possesses an environment that is highly reduced where both methanogenesis and sulfate reduction take place at extremely high rates. This was corroborated by the detection of a potent smell of hydrogen sulfide while measuring these extremely high CH4 fluxes. It is also possible that specific plant-soil relationships, such as higher polysaccharides in the

form of tree-root exudates (Lai, 2009) have enhanced CH4 production in the Tamarack zone. However, more research on the characteristics of the peat at this site is needed to reach conclusions about these extreme events.

Although higher heterogeneity in $CH_4$ fluxes within peat bogs can be encountered, it is likely that the same patterns of $CH_4$ flux along hydro-biological zones occur in other kettle-hole peat bogs due to the tight relationships between water level fluctuations and vegetation composition in these ecosystems (Malhotra et al., 2016). It is also possible that the higher rates of

$CH_4$ emission in this Ohio peat bog are replicated in similar peat bogs located at lower latitudes, where warmer temperatures have the potential to not only drive much higher productivity (Cai and Yu, 2011) but also increase methane emissions due to the effect of higher temperatures on $CH_4$ emissions in peatlands (Moore and Dalva, 1993; Pugh et al., 2018).

### 4.2. The role of plants in the CH4 cycle in peat bogs

The presence of different plant species was be strongly associated with variations in $CH_4$ emissions in peatlands. For example,

the presence of sedges, such as *Eriophorum vaginatum L.*, in ombrotrophic peat bogs was observed to be an important transport of $CH_4$ to the atmosphere (Greenup et al. 2000). In our study site, however, there was no active plant transport of $CH_4$. This lack of plant transport in ombrotrophic peat bogs has also been reported by Chasar et al. (2000) and can be likely attributed to a low abundance of sedges.

Lai et al (2014) found that fluxes varied significantly among plant communities at the ombrotrophic Mer Bleue bog in Canada.

In this bog, low fluxes were found in *Chamaedaphne* (32-22 mg $CH_4$ $m^{-2}$ $d^{-1}$) and *Maianthemum/Ledum* (83-53 mg $CH_4$ $m^{-2}$ $d^{-1}$) communities, whereas the highest were found in the *Eriophorum*-dominated community (122-124 mg $CH_4$ $m^{-2}$ $d^{-1}$). The magnitude of these fluxes was much lower than the average daily emissions from the mixed ericaceous shrubs of 224.72 mg $CH_4$ $m^{-2}$ $d^{-1}$.

Interestingly, we found that blueberry plants were slight but statistically significant sinks of $CH_4$. This result was also reported

by Sundqvist et al (2012), who found that boreal plants of spruce (*Picea abies*), birch (*Betula pubescens*), rowan (*Sorbus aucuparia*) and pine (*Pinus sylvestris*) showed a net uptake of $CH_4$. The values found by Sundqvist et al (2012) fluctuated between 1-2 nmol $m^{-2}$ $s^{-1}$, which is similar to the values found in this study. The mechanism behind this process is still uncertain but it has been reported that this process could be mediated by epiphytic bacteria capable of consuming $CH_4$ (Raghoebarsing et al., 2005). Sundqvist et al (2012) believe that the response is mediated by GPP and stomatal conductance through

mechanisms not yet understood.

We did not find a clear diurnal pattern of $CH_4$ emissions in the bog. Similarly, summer season measurements of eddy covariance in an ombrotrophic bog did not found clear diurnal patterns either (Nadeau et al., 2013). In contrast, studies in other wetlands have found a mid-morning peak in $CH_4$ emissions in fens (Whiting and Chanton, 1992) and marshes (Kim et al.,

1999; Rey-Sanchez et al., 2018; Van der Nat et al., 1998). This discrepancy is likely due to the fact that $CH_4$ emissions in marshes (Chu et al., 2014; Hatala et al., 2012; Morin et al., 2014, 2017), and in fens (Chasar et al., 2000; Treat et al., 2007; Waddington and Day, 2007), are largely dominated by plants that transport $CH_4$ through their aerenchyma.

### 4.3. Fluctuations in water level explain variability of $CH_4$ emissions

Methane fluxes were different among hydro-biological zones, but given that plants were not a pathway of $CH_4$ flux, the reported differences were most likely driven by the water level differences among hydro-biological zones. The length of dry conditions preceding permanently-wetted conditions has important consequences for the magnitude of $CH_4$ fluxes (Turetsky et al., 2014). While the highest $CH_4$ flux occurs after a period of 30 days of antecedent wet conditions (Turetsky et al., 2014), longer dry periods reduce the capacity of methanogens to acclimate to stable environmental conditions, therefore reducing

methanogenesis. Indeed, we found that the average water level data throughout 30 days prior to the flux measurement, not the instantaneous water level, had a significant effect in CH4 fluxes. We hypothesize that this is a general ecological response by which community composition lags behind environmental change. In our case, it may take several weeks for methanogens to acclimate to new water levels after the water level has been raised, therefore not responding to instantaneous changes in water level. Both Res and Shrubs zones were characterized by high fluctuations in water level, which was likely the cause of lower

$CH_4$ emissions in these zones when compared to the more permanently-wetted Tamarack, Mat and Water zones. Higher WL fluctuations in the Shrubs zones in 2018 (range: -40.4-6.1 cm) than in 2017 (range: -31.6-8.0 cm) could also explain the higher $CH_4$ emissions in 2017 than in 2018 in the Shrubs zone.

Our conclusion is that methanogen inhibition associated with longer dry periods in the Shrubs and Res zones is likely the cause of lower $CH_4$ emissions. However, reduced $CH_4$ emissions are also the result of an increase in the amount of methanotrophy

in the upper, oxic layers. We can confirm this as we observed pore-water concentration of $CH_4$ that were much higher in the Shrubs zone than in the Res zone, despite similar WL fluctuation. Yet, the fluxes were not significantly different between these two zones, indicating higher levels of methanotrophy in the Shrubs zones. Indeed, methanotroph relative abundance in the top section was twice as much in the Shrubs zone than in the Res zone.

We did not find a significant correlation between $CH_4$ flux and surface temperature. This is partially explained by the fact that

the effect of temperature on peatland $CH_4$ emissions is significant when the water table is near the surface (Strack and Zuback, 2013) and our site had significant water level fluctuations. For example, Lai et al. (2014) found that the relationship between temperature and $CH_4$ flux was only significant when the water table was less than 30 cm depth in average. It is possible that due to monthly variations in the water level in the Shrubs and Tamarack sites, the response of $CH_4$ emissions to temperature was confounded. The temporal resolution of the measurements was also a reason for the lack of correlation. At a higher

temporal resolution, such as the measurements of the diurnal pattern, the effect of temperature on $CH_4$ emissions may be more easily discerned.

## 4.4. Pore-water CH$_4$ concentrations were higher in the undisturbed section

Pore-water CH$_4$ concentration was high throughout the undisturbed section of the bog and significantly lower in the restored section. Although concentrations of key electron acceptors, such as nitrates or sulfates, were low and not significantly different among zones, we found that the restored section had significantly higher concentration of Mn (F = 3.80, p = 0.01) and Na (F = 3.78, p = 0.01), suggesting bacterial manganese reduction could compete against methanogens in the restored zone.

Excluding the Mat zone, pore-water CH$_4$ concentration followed a similar pattern of variation to the fluxes, with higher concentrations in the Tamarack zone, followed by Shrubs, Res and Lagg zones. Low concentration but higher fluxes in the Mat zones indicate a higher CH$_4$ transfer velocity. This could be the consequence of different porosities in the peat that affect the rate of transfer. However, because the porosity throughout the peat bog was uniform, it is likely that CH$_4$ transfer velocity is being driven by microbial activity rather than physical properties (see section 4.6.).

Pore-water CH$_4$ concentration was the highest in the Tamarack zone, with concentration at deeper levels close to the saturation point (1.2 mM). Similarly, in a study in an ombrotrophic peat bog in Minnesota, Chasar et al. (2000) reported high CH$_4$ pore-water concentrations in bogs of 1.2 and 1.5 mM for porewater at about 1m of depth for June and July, respectively. Chasar et al. (2000) also reported much higher pore-water CH$_4$ concentrations in bogs than in fens, and suggested that this is related to negligible plant transport in peat bogs that causes CH$_4$ to accumulate in the porewater, diffuse upwards and be oxidized in the top layers of the peat. Methanotrophy in the shallow layers of the peat was also reported by Chasar et al (2000), where analysis of isotopes in shallow pore water versus associated fluxes, indicated oxidation of CH$_4$ in the pore-water before diffusive transport to the atmosphere.

Concurrent measurements of pore-water CO$_2$ concentrations indicated that the CH$_4$:CO$_2$ ratio was similar at the top of the profiles, while at the bottom of the profiles there was a clear difference between restored and undisturbed sites (Appendix B, Fig B3). This difference could indicate that there is a higher competition for respiratory processes in the disturbed section, while methanogenesis is more favored in the undisturbed section. The analysis of CO$_2$ fluxes is not, however, within the scope of this manuscript and are here presented only as a preamble for future studies.

## 4.5. Methane-cyclers abundance depends on vegetation zone and water level

Consistent with expectations based on their anaerobic lifestyle, we found higher relative abundances of methanogens in the permanently-wetted areas Mat and Tamarack, than in the intermittently-wetted areas (Shrubs, Res-N and Res-S). Hydrogenotrophic methanogens, which are typically dominant in nutrient-poor sites (Kelly et al., 1992; Kim et al., 2008; Kotsyurbenko et al., 2004) and are typical of Sphagnum-dominated bogs (Chasar et al., 2000; Kelly et al., 1992; Lansdown et al., 1992), dominated both the undisturbed and restored sections, while acetoclastic methanogens were rare and only slightly more common in the restored section. We hypothesized that the restored section had gained more nutrients due to higher degree of mineralization, however, the dominance of hydrogenotrophic methanogens suggests that the restored section may still be nutrient-poor, despite the disturbance and apparent mineralization of the soil. This is also evident by the low concentration of

key constituents, such as nitrates, iron, ammonium, phosphorous and magnesium (although note relatively higher concentrations of manganese and calcium in the restored section; Appendix A, Table A2). It is possible that 15 years of restoration efforts have effectively restored this section's trophic status and that acetoclasty was higher there in the past. Alternately, the original disturbance may have had minimal impact on the microbial composition, such that the restored section retains a community similar to its pre-disturbance state, when it was part of the Shrubs zone of the then-undisturbed section. Basiliko et al. (2013) similarly found that mining-based disturbance and subsequent restoration of Canadian peatlands did not affect archaeal microbial community composition.

At the genus level, however, there were differences in methanogen composition between the undisturbed and restored sections. While hydrogenotrophic genera strongly dominated both, there was a shift from *Methanoregula*-dominated communities in the undisturbed sections to strongly *Methanobacterium*-dominated communities in the restored sections. Based on our prediction of higher nutrient status in the restored site, we would have expected the opposite trend in *Methanoregula* dominance, since *Methanomicrobiales* (the order containing *Methanoregula*) have been observed to prefer nutrient-rich sites (Godin et al., 2012); their dominance is further indication that the restored section is not as high-nutrient as we expected. In contrast to the hydrogenotrophs, the acetoclasts did not show genus-level differences from undisturbed to restored zones, but rather from inundated (Mat-S and Tamarack-S) to intermittently flooded (Shrubs, Res-N and Res-S) ones. When acetoclasts were present, *Methanosaeta* dominated their community, consistent with observations of *Methanosaeta* in nutrient-poor acidic sites (Godin et al. 2012). However, in the inundated zones, *Methanosarcina* was also present. This is actually the opposite pattern we would have expected based purely on likely oxygen concentrations, as Methanosaeta typically dominates anaerobic environments while Methanosarcina can produce methane under partially oxic conditions (Angel et al., 2011). We therefore interpret Methanosaeta's presence in FSL-S and TMW-S to arise from its greater metabolic versatility – in addition to acetate, it can also use CO2 or methylated compounds (Liu and Whitman, 2008) – and thus that these sites may have distinct substrate profiles

Methanotrophic lineages, like methanogens, were at the highest relative abundances in the undisturbed, inundated sites, where they primarily occurred near the peat surface. The higher abundance of methanotrophs in inundated zones may be related to the presence of *Sphagnum* mosses in these zones as methanotrophs are a common, abundant member of the *Sphagnum* microbiome (Dedysh, 2011; Kostka et al., 2016); the DNA extraction method may have accessed microbiota on and within the moss as well as from the bulk peat. Alternatively the higher methanotrophs in the inundated sections may have been in the bulk peat, and simply be due to the higher supply of methane in those areas.

### 4.6. Microbiota drive CH₄ transfer velocity

While methanogens control the production of $CH_4$ through the peat column, methanotrophs interact with plants and physical processes to mediate what portion of produced $CH_4$ is oxidized before being emitted to atmosphere. We therefore examined the relationship between resident $CH_4$ cyclers and the $CH_4$ transfer velocity in the top stratigraphic layer of the peat. Generally, when the water level is near the surface, $CH_4$ diffuses directly from the surface pore water to the overlying air and can also be

transported via plant tissue. However, at our site, we measured no significant $CH_4$ transport through vascular plants (see section 4.2). Therefore, the transport pathway at the upper layers of the soil in all zones (except Water), should occur through the ubiquitous *Sphagnum* mat and thus have similar resistance throughout the site. We also found no significant correlation between $CH_4$ pore-water concentration at the top soil layers and surface $CH_4$ flux (Fig. 7a). Thus, with all zones expected to

be similar in their physical transport processes, observed differences in $CH_4$ transfer velocity among zones should represent differences in microbial processes. Indeed, we found a significant correlation between $CH_4$ transfer velocity and the ratio of total methanogens to total methanotrophs ($r^2 = 0.33$, $p = 0.03$ based on the relative abundances of lineages in each functional guild, see Methods) (Fig. 9). A related result was reported in a rice paddy system, where the ratio of the gene expression of the two diagnostic marker genes for methanogenesis and methanotrophy, *mcrA* and *pmoA*, in the upper 10cm of the soil was highly

correlated to $CH_4$ flux (Lee et al., 2014). However, in our site, this correlation was not significant when using $CH_4$ flux data alone ($r^2 = 0.01$, $p = 0.75$) or pore-water data alone ($r^2 = 0.03$, $p = 0.57$). It is intriguing that, despite the fact that presence does not necessarily imply activity, and relative abundances do not represent absolute abundances, in our study we see this relationship between the 16S rRNA gene amplicons of known methanogenic and methanotrophic lineages and the $CH_4$ fluxes in both undisturbed and restored peatlands. This result illustrates the utility of examining microbiota to explain differences

between $CH_4$ production and emissions to the atmosphere.

## 5. Conclusions

Flatiron Lake Bog had high rates of $CH_4$ emission that included several non-ebullitive extreme fluxes occurring in the Tamarack-mixed woodland zone. $CH_4$ emissions decreased with distance from the center of the bog, from regularly-wetted sections to those that had higher water level fluctuations. Pore-water concentrations followed a similar pattern of increase with

depth, except for the Mat zone, which is adjacent to the open water and thus has better vertical mixing. Longer dry periods in the Shrubs and Res zone likely inhibited methanogens, lowering their abundance and thus decreasing $CH_4$ accumulation in the pore-water and associated emissions. Although pore-water chemistry explained some of the variation in pore-water $CH_4$ concentration, water level explained the largest component of variation in $CH_4$ fluxes due to its effects on methanogenesis and methanotrophy at the top soil levels. Given that plants were not an appreciable pathway of $CH_4$ flux, the reported differences

in $CH_4$ transfer velocity when the water level was high were explained by the ratio of the relative abundance of methanogens to methanotrophs in the top layer.

Why would two locations with similar near-surface $CH_4$ concentrations have different fluxes if they also have similar diffusivities and negligible ebullition and plant transport? Our results show the answer is that they have different transfer velocities for $CH_4$. Transfer velocities are normally a function of wind speed, but beneath the shrub and tree canopy of peat

bogs wind speeds are very low so something else is affecting this transfer velocity. The upper layer of the bog's peat mass is a dynamic region with both methanotrophs and methanogens living within the oxic layer (Angle et al., 2017). Within this layer higher abundance of methanogens drive higher transfer velocities if the concentration of $CH_4$ is assumed to be at quasi-steady

state. At the same time, however, methanotrophs consume much of the methane produced. Therefore, methanogen abundance, when normalized by methanotroph abundance, can explain $CH_4$ transfer velocity differences in a peat bog where diffusive transport from porewater in saturated layers is dominant. We conclude that microbial communities, and their control by variation in water table depth, are the key drivers of variability in $CH_4$ fluxes across multiple hydro-biological zones in kettle-

hole peat bogs. Future research should examine whether such patterns can be confirmed in other ecosystems where plant-mediated transport of $CH_4$ is low.

## Author contributions

CRS, GB and GMD designed the experiments. CRS, JS, RGA, and YH conducted field and laboratory observations. VR and
YFL generated the microbial analyses. CRS prepared the manuscript with contributions from all co-authors.

## Competing interests

The authors declare that they have no conflict of interest.

## Code/Data Availability

The data and the code used in this manuscript are available upon request.

## Acknowledgements

We thank Bryan Cassidy, Dominique Hadad, Anna Thompson, Julio Quevedo, Austin Rechner, and Alexa Baratucci for their assistance in the field in 2017, and Tasmina Uddin, Tim Becker, Charles Davis, Jorge Villa, Theresia Yasbeck, Taylor Stephen,
Yang Ju, and Cassandra Rey for their assistance in the field in 2018. We thank Julian Deventer for insights on the analysis of the data. We also thank Karen Seidel and The Nature Conservancy for granting access to the site. This work was supported by the Presidential Fellowship at The Ohio State University awarded to CRS, and by the Ohio Water Resources Center project G16AP00076, OSU OARDC award 2016-055 SEEDS, and the OSU Office of Energy and Environment.

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

**FIGURES**

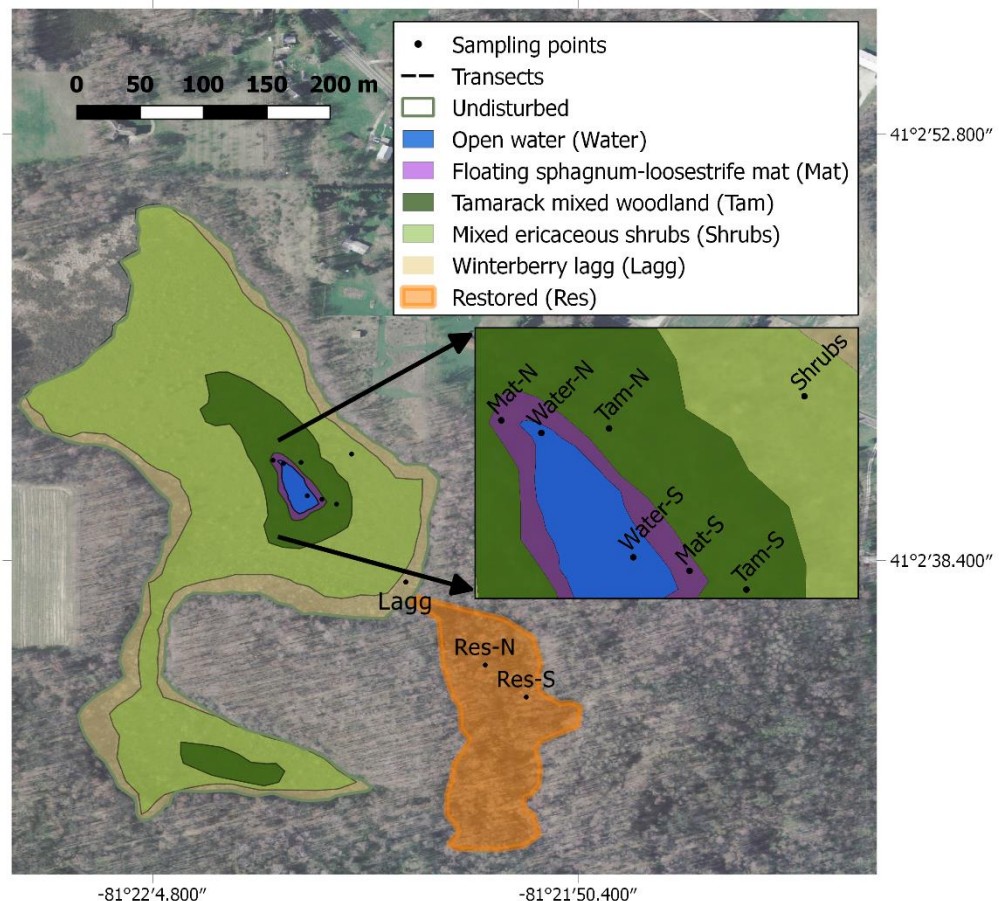

Figure 1. Map of the study site showing the different hydro-biological zones, and the sampling locations.

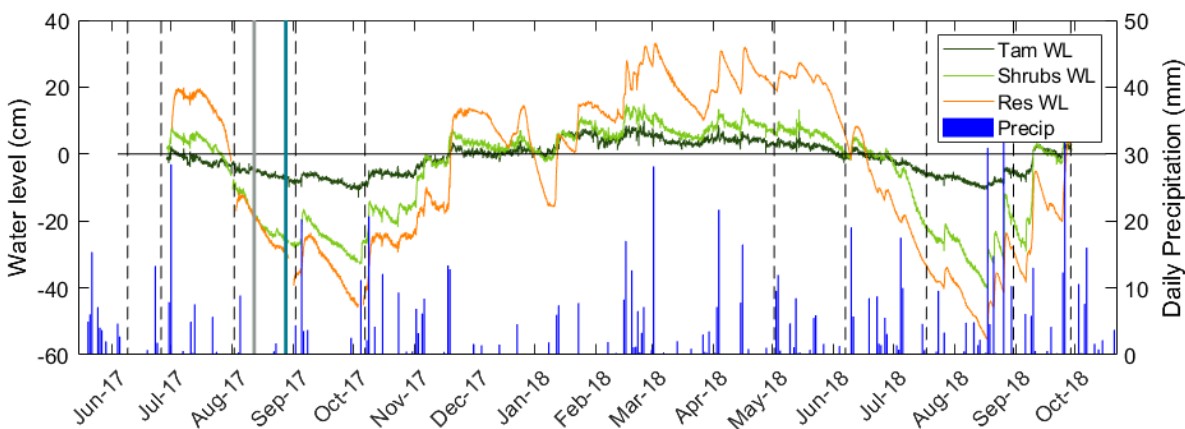

Figure 2. Water level (WL) fluctuations in the Tamarack Mixed Woodland (Tam) zone, the Mixed Ericaceous Shrub (Shrubs) zone and the restored (Res) zone of the bog. Vertical dashed lines indicated the ten times of pore-water sampling, and the solid lines indicate the two times of core sampling: gray for Tam, Shrubs, and Res-S, and teal for Mat and Res-N. The secondary

10   axis shows daily values of precipitation.

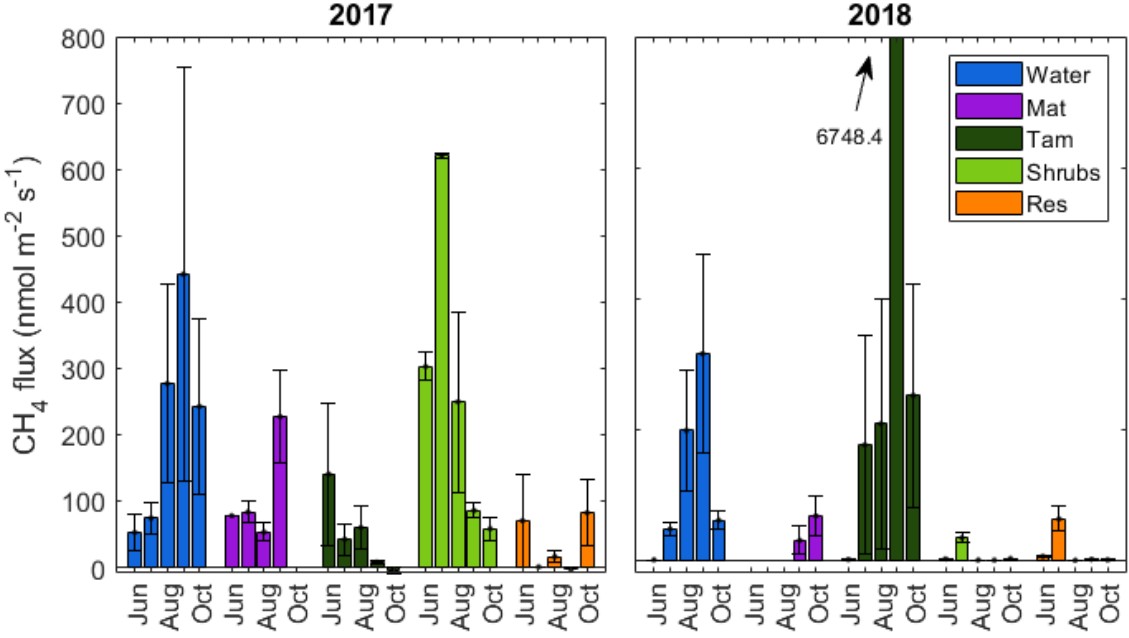

Figure 3. Monthly fluxes of methane for each of the five hydro-biological zones of the study. Fluxes from the Lagg were not significantly different than zero and are therefore not shown. Standard errors are for all sample locations within the same

5    month and zone (variable number, see section 2.3)

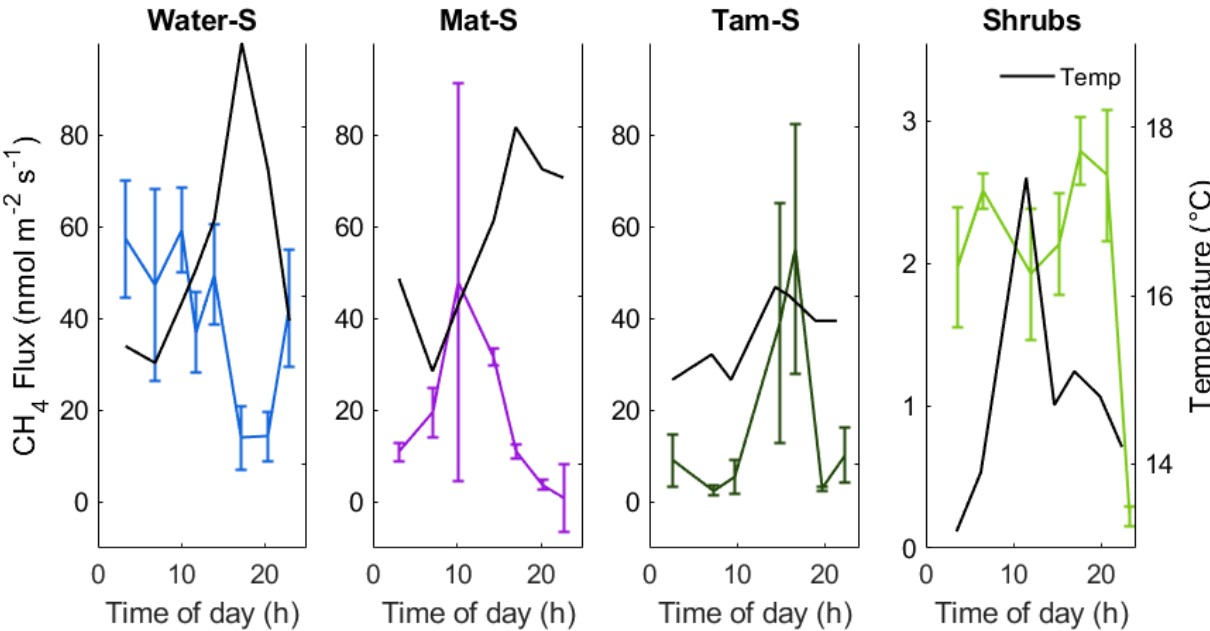

Figure 4. Diurnal patterns of $CH_4$ emission measured over a 24-hour period in September 2018. Note a smaller y-axis maximum in the Shrubs zone. Error bars represent the standard error of 4 individual chamber measurements within the same 30-minute period at each location. Secondary axis (and black lines) shows the temperature at 10 cm below the surface either in the open water (Water) or in the peat (Mat, Tamarack, and Shrubs). The Res zone was not sampled.

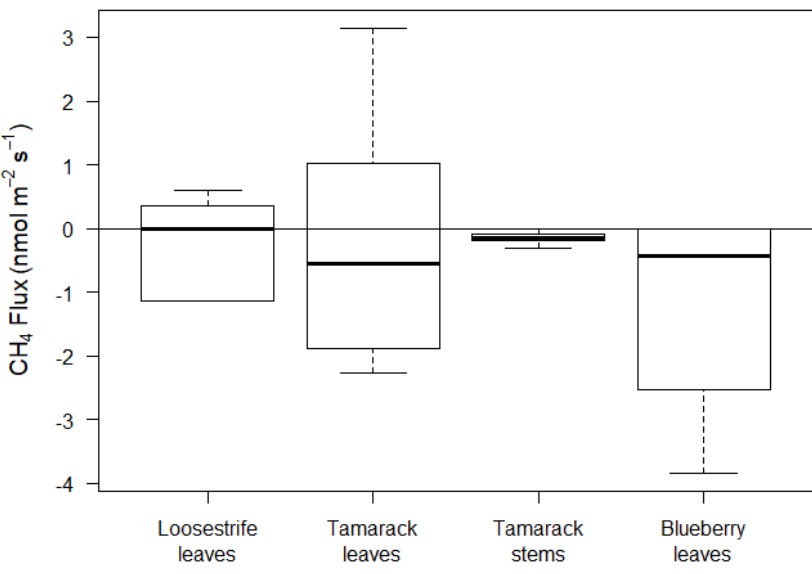

Figure 5. Plant-mediated CH$_4$ fluxes, from loosestrife leaves (Mat zone), tamarack leaves, tamarack Stems, and blueberry leaves (Tamarack and Shrubs zone). Only fluxes from the blueberry were significantly different from zero (p = 0.01).

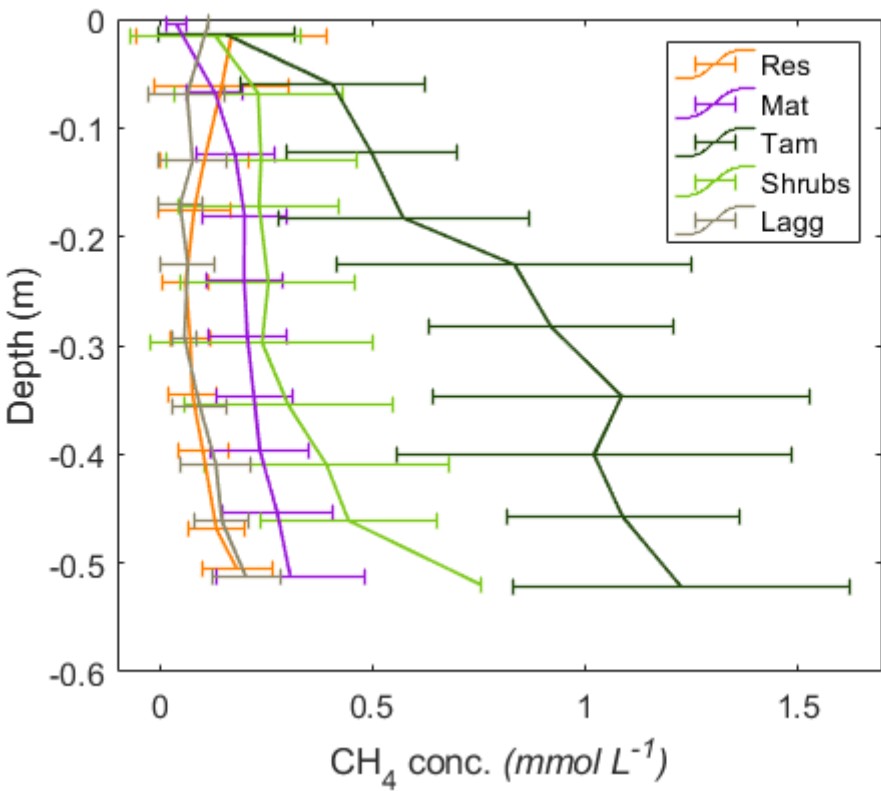

Figure 6. Vertical profiles of $CH_4$ pore-water concentrations by zone. The error bars represent the standard deviation of the monthly measurements for 2017 and 2018, combined. A minor y-axis jitter has been added to more clearly distinguish zone patterns. Note that the concentrations in the Tamarack zone at depth approach saturation (1.44 mM, at 20 °C $CH_4$).

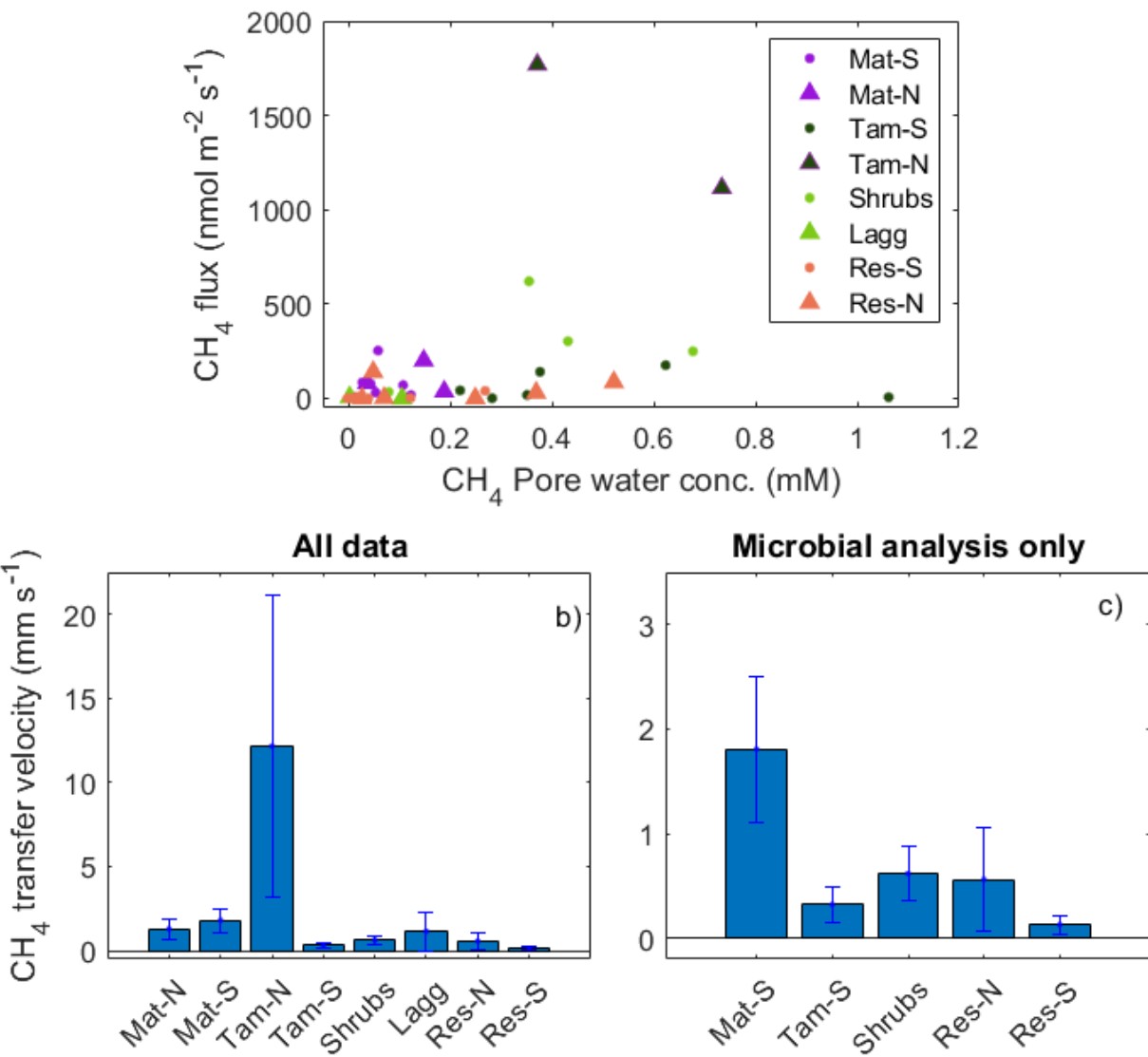

Figure 7. a) Relationship between $CH_4$ pore-water concentration and $CH_4$ flux for times where the WL was high and within the top stratigraphic layer of the peat. b) $CH_4$ transfer velocity calculated from the upper plot. c) same as previous but with the data relevant for microbial analysis only. Note that microbial samples for Tam-N, Mat-S, and Lagg are not available and therefore not used in the following comparisons of $CH_4$ transfer velocity against microbial activity. The error bars are the standard error.

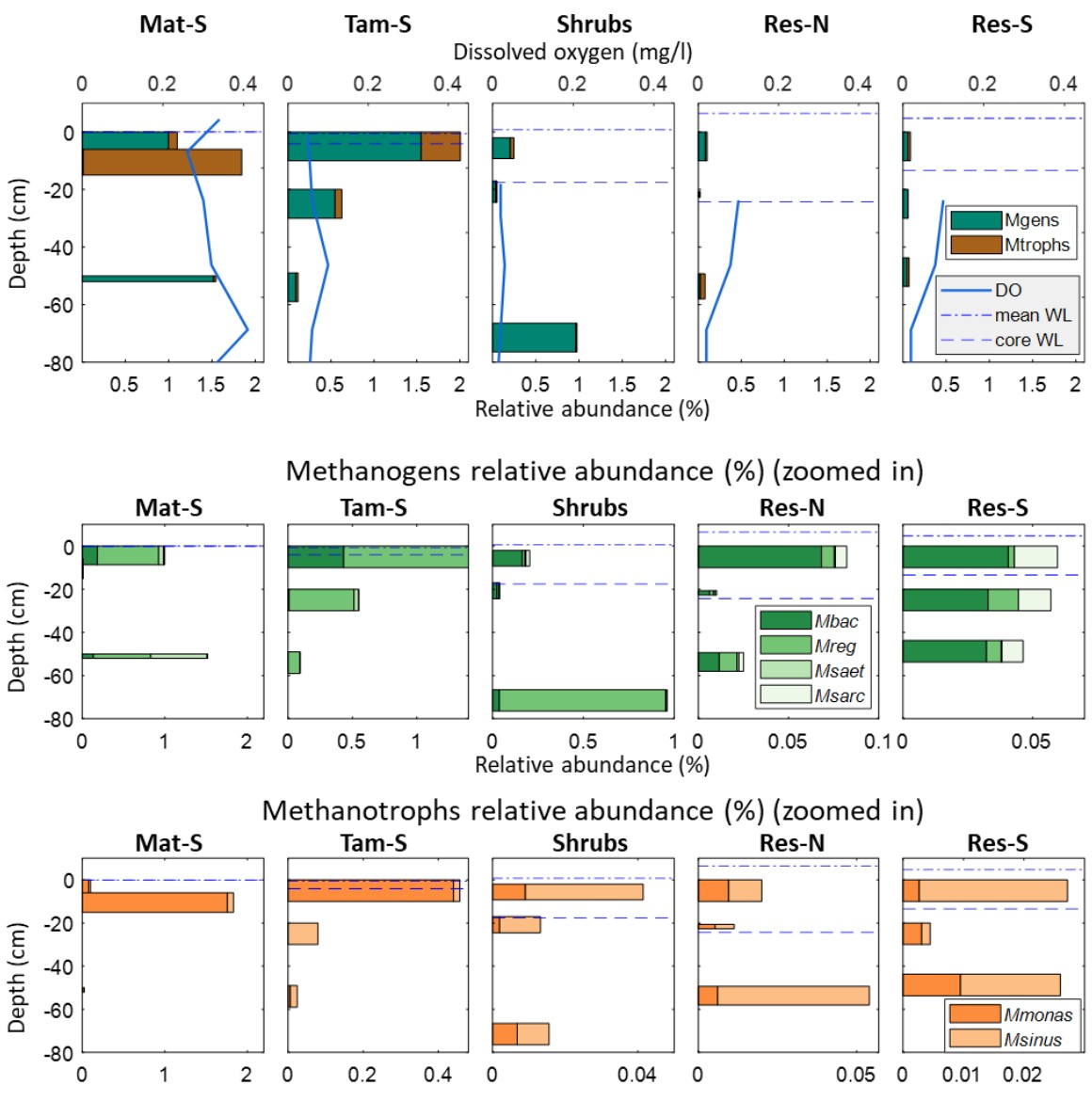

Figure 8. Relative abundances of methanogens and methanotrophs in the Mat-S, Tam-S, Shrubs, Res-N, and Res-S zones of the bog, at different depths in the peat column, with the mean water level from June 2017 through August 2017 (mean WL) and the water level at time of sampling (core WL) (in Mat-S these were both at 0cm; in Tam-S, the mean WL was at 0cm).

The upper panel shows overall methanogen ('Mgens') and methanotroph ('Mtrophs') abundances, along with the average dissolved oxygen profile over the preceding month (from coring; see Methods).  The observed genera of methanogens and methanotrophs are shown on the middle and lower panels, respectively, with variable x-axes. *Methanobacterium* (Mbac) and *Methanoregula* (Mreg) are hydrogenotrophic methanogens, and *Methanosaeta* (Msaet) and *Methanosarcina* (Msarc) are acetoclastic methanogens. *Methylomonas* (Mmonas) and *Methylosinus* (Msinus) are methanotrophs.

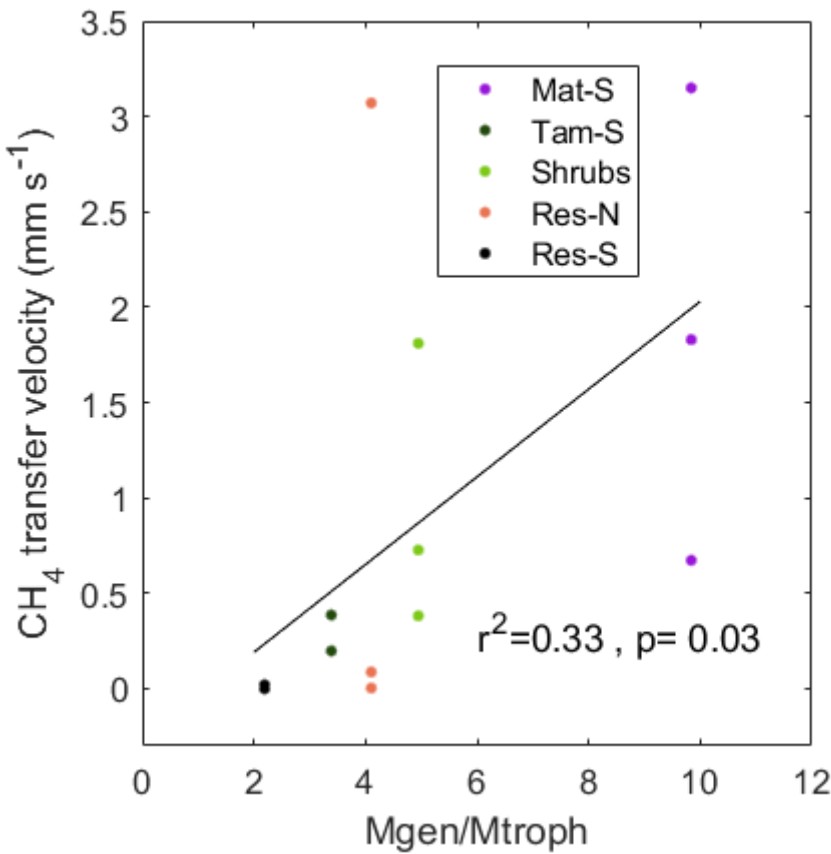

Figure 9. Relationship between the ratio of the relative abundance of methanogenic/methanotrophic lineages ('Mgen/Mtroph') and the CH$_4$ transfer velocity in the top stratigraphic layer of the peat profile: 0-6.7 cm for Mat, and 0-12.5 cm for the other zones. CH$_4$ transfer velocity was calculated as the average for the three months prior to coring, during which the water level was within or above the top stratigraphic layer.

Table 1. CH$_4$ fluxes for the different hydro-biological zones in Flatiron Lake Bog. Integrated fluxes are based on a 122-day period for 2017 and 149-day period for 2018. Values in parenthesis for mean fluxes are the standard error and for the subsequent rows the propagated standard error.

| | Area (m2) | Median flux (nmol m$^{-2}$ s$^{-1}$) | Mean flux (nmol m$^{-2}$ s$^{-1}$) | Daily Normalized emissions 2017 (mg CH$_4$ m$^{-2}$ d$^{-1}$) | Daily Normalized emissions 2018 (mg CH$_4$ m$^{-2}$ d$^{-1}$) |
|---|---|---|---|---|---|
| **Water** | 1119 | 61.9 (0.1–1823) | 122.6 (24.5) | 349.2 (402.5) | 210.7 (169.8) |
| **Mat** | 927 | 39.5 (-12.2–753) | 82.5 (20.7) | 154.1 (86.9) | 83.2 (48.5) |
| **Tamarack** | 14577 | 10.5 (-8.6–27186) | 602.2 (342.1) | 68.6 (82.8) | 2478.5 (3819.5) |
| **Shrubs** | 85539 | 3.1 (-0.8–624) | 62.5 (22.62) | 441.0 (162.7) | 8.5 (9.0) |
| **Res** | 23430 | 0.7 (-11.4–279.9) | 21.5 (8.18) | 30.3 (50.8) | 18.8 (27.5) |
| **BB** | 123546 | -0.4 (-3.8– -3.84) | -1.1 (0.47) | -3.9 (1.8) | -3.8 (1.3) |
| **Total** | 125592 | NA | NA | 315.4** (166) | 362.3** (687) |

* BB: Blueberry leaves occupy the area of the Tamarack, Shrubs and Res zones. Fluxes from other plant species and from the Lagg zone were not significantly different from zero

** Total emissions per zone (mg CH$_4$ d$^{-1}$) were added and divided by the area of the bog (excluding the Lagg zone) to produce the final result of 315.4 ± 166 mg CH$_4$ m$^{-2}$ d$^{-1}$ in 2017 and 362.3 ± 687 mg CH$_4$ m$^{-2}$ d$^{-1}$ in 2018.

**Appendices**

**Appendix A**

Table A1. subset of amplicon-based lineages identified as genera of known methanogens and methanotrophs The genera found in the study are shown in bold letters.

| Methanogens | Methanotrophs |
|---|---|
| **Methanobacterium** | Methylocystis |
| Methanobrevibacter | **Methylosinus** |
| Methanocalculus | Methylocella |
| Methanocaldococcus | Methylocapsa |
| Methanocella | Methyloferulla |
| Methanococcoides | Methylococcus |
| Methanococcus | Methylocaldum |
| Methanocorpusculum | Methylomicrobium |
| Methanoculleus | Methylosphaera |
| Methanofollis | **Methylomonas** |
| Methanogenium | Methylobacter |
| Methanohalobium | Methylosarcina |
| Methanohalophilus | Methylothermus |
| Methanolacinia | Methylohalobius |
| Methanolinea | |
| Methanolobus | |
| Methanomassiliicoccus | |
| Methanomethylovorans | |
| Methanomicrobium | |
| Methanomicrococcus | |
| Methanoplanus | |
| Methanopyrus | |
| **Methanoregula** | |
| **Methanosaeta** | |
| Methanosalsum | |
| **Methanosarcina** | |
| Methanosphaera | |
| Methanosphaerula | |
| Methanospirillum | |
| Methanothermobacter | |
| Methanothermococcus | |
| Methanothermus | |

Methanotorris

Methermicoccus

Methanoflorens

Methanomassilliicoccus

Methanospaerula

Methanospirillium

Methanothrix

Table A2. Water chemistry of the pore water in the eight locations of the study. The means (SD) are averages of four measurements taken throughout the growing season of 2017. Asterisks indicate means that are significantly higher than at least another mean.

| Variable | Mat-N | Mat-S | Tam-N | Tam-S | Shrubs | Lagg | Res-N | Res-S |
|---|---|---|---|---|---|---|---|---|
| NH4 (mg $L^{-1}$) | 0.53 (0.24) | 0.47 (0.09) | 3.5 (0.75)* | 2.85 (0.13)* | 1.22 (0.42) | 0.54 (0.55) | 1.62 (1.65) | 1.36 (0.17) |
| NO3 (mg $L^{-1}$) | 0.06 (0.03) | 0.03 (0.01) | 0.08 (0.04) | 0.03 (0) | 0.08 (0.09) | 0.04 (0.01) | 0.07 (0.08) | 0.05 (0.03) |
| pH | 4.1 (0.36) | 4.7 (0.43) | 4.26 (0.19) | 4.88 (0.5) | 4.75 (0.58) | 4.32 (0.3) | 5.38 (0.82)* | 4.42 (0.42) |
| EC (dS $m^{-1}$) | 0.04 (0) | 0.04 (0) | 0.08 (0) | 0.08 (0.03) | 0.06 (0) | 0.07 (0.02) | 0.09 (0.04)* | 0.08 (0.01) |
| Al (mg $L^{-1}$) | 0.57 (0.07) | 0.47 (0.07) | 0.64 (0.08) | 0.18 (0.07) | 0.27 (0.11) | 1.07 (0.28)* | 0.34 (0.25) | 0.79 (0.35) |
| Ca (mg $L^{-1}$) | 3.1 (0.36) | 3.15 (1.03) | 4.4 (0.62)* | 1.46 (0.5) | 2.55 (0.65) | 4.52 (1.17) | 4.39 (1.45)* | 4.16 (1.59)* |
| Fe (mg $L^{-1}$) | 1.32 (0.14) | 1.28 (0.16) | 1.42 (0.13) | 0.76 (0.67) | 1.46 (0.91) | 2.01 (0.78) | 0.97 (0.59) | 0.82 (0.27) |
| K (mg $L^{-1}$) | 1.56 (0.27) | 2.05 (0.51) | 2.38 (0.45) | 2.42 (0.28) | 2.38 (0.25) | 9.41 (10.83) | 2.06 (0.62) | 9.29 (12.63) |
| Mg (mg $L^{-1}$) | 0.91 (0.09) | 0.85 (0.1) | 1.38 (0.11) | 0.53 (0.11) | 0.91 (0.08) | 1.24 (0.27) | 1.09 (0.44) | 0.89 (0.27) |
| Mn (mg $L^{-1}$) | 0.05 (0) | 0.05 (0) | 0.11 (0.01)* | 0.04 (0.01) | 0.07 (0.01) | 0.09 (0.03)* | 0.1 (0.01)* | 0.1 (0.03)* |
| Na (mg $L^{-1}$) | 1.07 (0.33) | 1.1 (0.31) | 1.55 (0.21) | 1.69 (0.28) | 1.55 (0.17) | 2 (0.77) | 2.5 (1.64) | 2.09 (0.34) |
| P (mg $L^{-1}$) | 0.07 (0.02) | 0.04 (0.01) | 0.16 (0.04) | 0.04 (0.02) | 0.06 (0.06) | 0.1 (0.06) | 0.07 (0.05) | 0.33 (0.16)* |

| | | | | | | | | |
|---|---|---|---|---|---|---|---|---|
| S<br>(mg L$^{-1}$) | 1.84<br>(0.43) | 1.64<br>(0.27) | 4.95<br>(0.47)* | 1.71<br>(0.37) | 1.92<br>(0.43) | 2.31<br>(0.53) | 2.37<br>(1.28) | 2.82<br>(0.59) |
| Zn<br>(mg L$^{-1}$) | 0.12<br>(0.07) | 0.23<br>(0.2) | 0.26<br>(0.32) | 4.06<br>(7.76) | 4.7<br>(4.61) | 0.31<br>(0.21) | 11.38<br>(12.57) | 4.04<br>(6.03) |

* level of significance $p < 0.05$

**Appendix B**

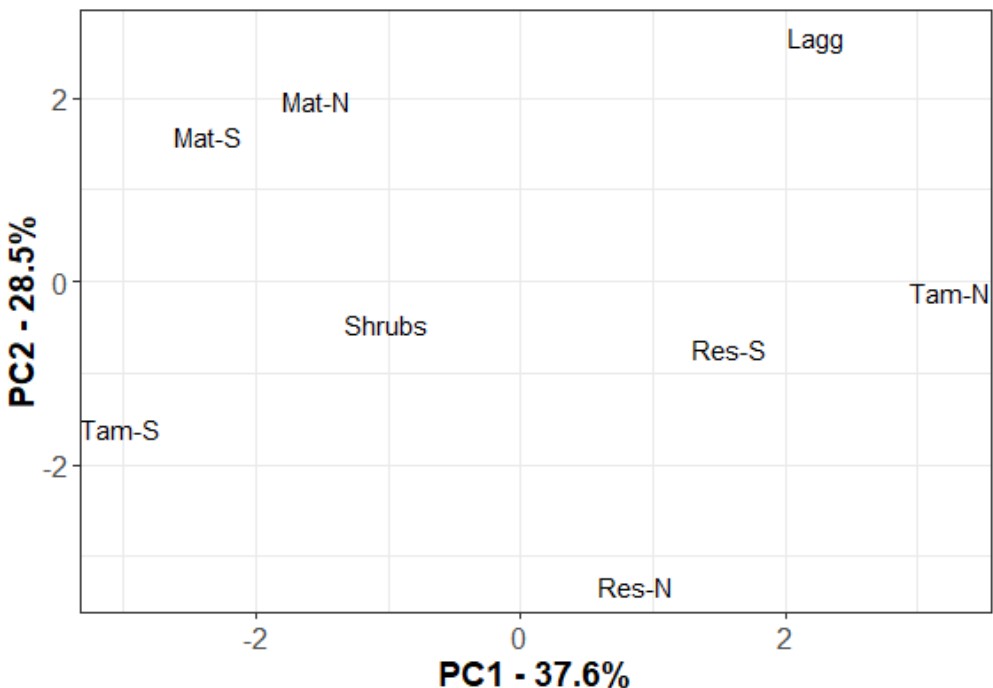

Figure B1. Principal Component Analysis of the 14 variables listed in Table A2.

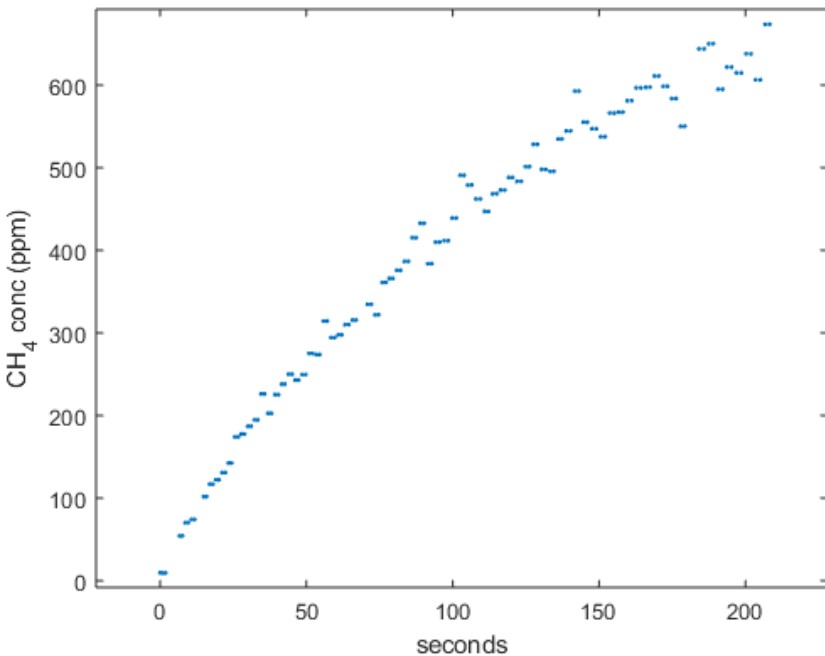

Figure B2. Chamber measurement during the September hotspot in the Tam-N location. Note the steady increase in concentration that indicates that ebullition was not the reason for the high magnitude of the flux at this location.

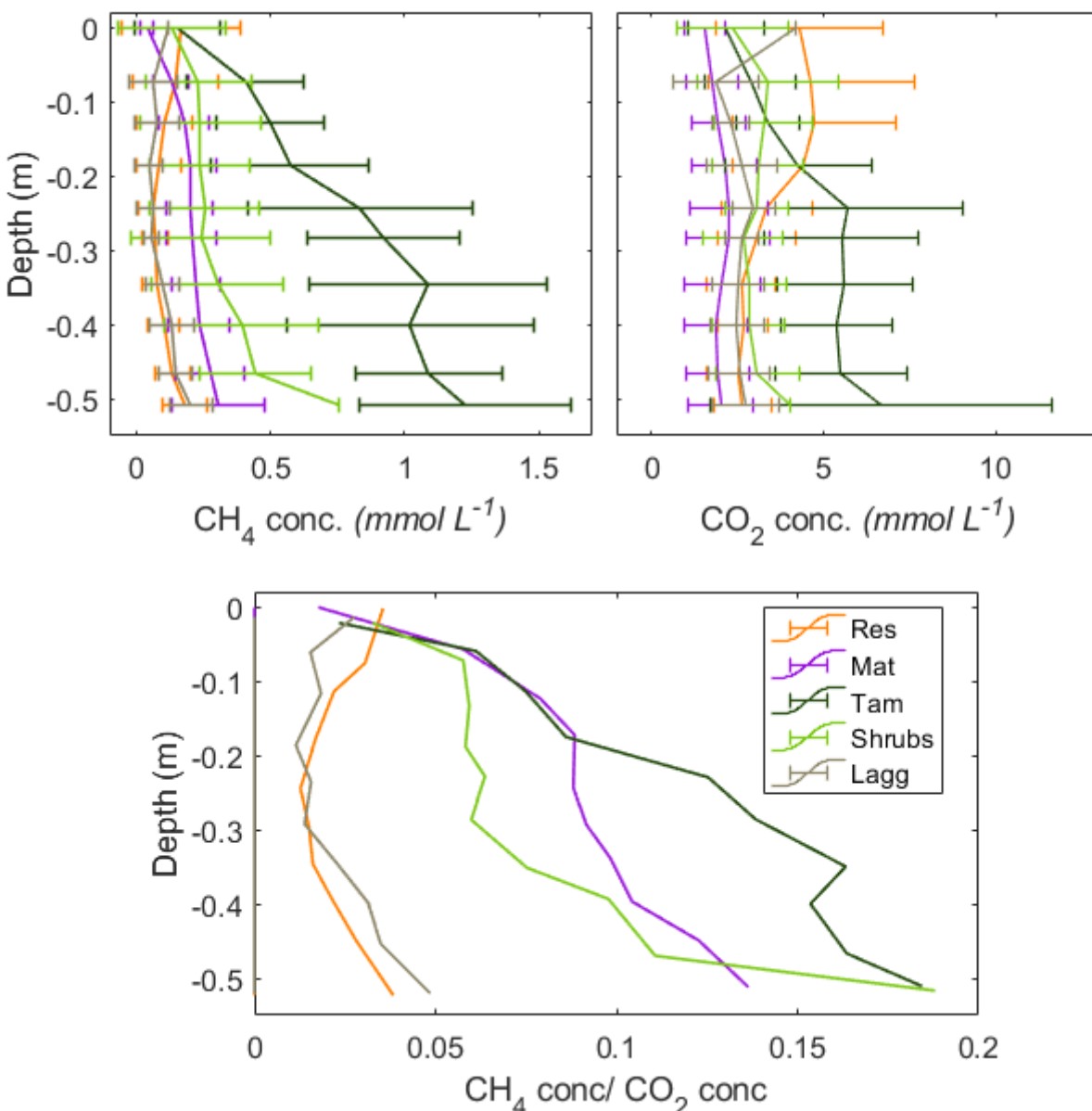

Figure B3. Vertical profiles of CH$_4$ and CO$_2$ pore-water concentrations (top) and the resulting CH$_4$:CO$_2$ ratios (bottom).