# Peer review of "The ratio of methanogens to methanotrophs and water-level dynamics drive methane transfer velocity in a temperate kettle-hole peat bog"

_Biogeosciences, 2019_

## Referee Comment (RC1) · Anonymous Referee #1 · 4 Jun 2019

The authors study methane fluxes and their drivers in a kettle bog in Ohio in 2017 and 2018. They present monthly chamber fluxes and porewater concentration profiles from transects starting at the central open water area and ending at the upland. In addition, they sampled leaf level CH4 fluxes in 2018 and took cores to determine the relative abundance of methanogens and methanotrophs (via 16s rRNA sequencing). Their objectives are to estimate the growing season budget and to analyze biotic/abiotic drivers as well as the relationship between microbial population and the observed fluxes. The study site incorporates a restoration site, which has also been sampled and differences between undisturbed and restored sites are discussed. They find temporally highly variable fluxes and the integrated budget indicates rel. high fluxes compared to other boreal bogs (possibly due to the location at the southern border of the boreal zone, and thus higher temperatures). They find water table level a month prior to flux sampling as most important driver. Both methanogens and methanotrophs are most frequent at sites that are permanently water saturated. The authors calculate a 'methane exchange velocity' based on top soil methane concentration and the measured fluxes and can correlate this with the ratio of methanogens and methanotrophs (rel. abundance). They conclude that microbes and intermittency of the water table are important drivers for methane cycling in bogs. Methane fluxes are variable in space and time. To a large degree, the associated uncertainties come from dynamics related to the two microbial processes involved (methane production and oxidation) as well as the different transport mechanisms (diffusion, plant-mediated, ebullition). Often, water table level is used as a rough threshold between anaerobic (i.e. methane producing zone) and aerobic (i.e. methane oxidation zone). Thus, water table has often been found as important driver in small-scale flux studies. To me the most interesting part of the study is the attempt to link fluxes and microbial populations and I am intrigued by the approach to estimate the 'methane exchange velocity'. However, I think that this approach needs a more theoretical base and a discussion of its possible advantages and limitations. 1) Methane exchange velocity: The authors do not give any information about the assumptions that go in equation (1). It seems that a) ebullition and plant-mediated transport have to be excluded and b) the peat structure and water/air content has to be the same for all sites (i.e. diffusivity is identical as well). Thus, by default, the only remaining factor to explain fluxes is the net methane production (i.e. microbial processes). And that is indeed, what the authors find. Only after reading the whole manuscript, it becomes clear that assumption a) is fulfilled (although the high fluxes in summer 2018 are unexplained). 2) Microbial populations and activity: The author correctly state, that their analysis only indicates the presence of microbes, not their activity (i.e. gene expression, as was done in the Lee 2014 paper, which is cited

here). However, this makes the interpretation of Fig. 8 more difficult. I would like to point out FSL-S: Fig. 7 shows that at FSL-S, very close to the top soil, methanotrophs dominate. But for the relation with 'methane exchange velocity', only top soil ratio of methanogens/methanotrophs is used (Fig. 8) – where FSL with both high ratio of methanogens/methanotrophs and high (but variable) methane exchange velocity is clearly needed for the correlation. Given these assumptions, I wonder whether the monthly porewater concentration profiles (Fig. 6 only shows the overall mean profiles) contain more information about production, oxidation and diffusive transport (i.e. the shape of the profile). If so, this can be used as further support of microbial activity as most important driver. Minor comments Page 7: Considering the high fluxes in summer 2018, I wonder whether the starting point may already have been high (i.e. an ebullition event early on)? Could be helpful to include the graph in the Appendix. Page 9: I understand why and how you do the up-scaling of chamber fluxes. However, there really is no way of evaluating that number and given the temporal variability there is the possibility that the large integrated flux is due to that (but temperature as discussed is possible as well). Page 17: Is there an explanation for the result, that the instantaneous water table does not have a significant effect, but the one a month earlier has? Page 19: Given the importance of the methane exchange velocity, I would move the figure from the appendix up to the main text and also discuss its error (from the figure it looks like that only for FSL and TMW the estimate is significant?).

―――――――――――――――――――――――――

---

## Referee Comment (RC2) · Anonymous Referee #2 · 4 Jun 2019

The study reports methane (CH4) fluxes over two growing seasons (2017 & 2018) from a kettle-hole peat bog and captures spatial (vegetation zone) and temporal (monthly) variability with its sampling design, as well as the fluxes associated with plant stems and leaves. The authors also report dissolved CH4 profiles and soil biogeochemical and microbial community characteristics.

The manuscript is well-written, clearly presented, and provides a thorough methods description, though at times the phrasing could be briefer. The flux data are valuable given that the system represents a warm end-member for temperate bogs and the

slight CH4 sink for some of the plant stem observations is interesting. The attempt to link biogeochemical process (CH4 flux) to microbial community data is also a strength of the study, especially the breakdown of lineages of methanotrophs and methanogens by genus, however the limitations should be more carefully discussed, for example that DNA is not indicative of activity.

The weakness of the study is that the temporal coverage and frequency of flux observations is relatively scarce despite the well-known high variability associated with methane flux. The latter means both that drivers such as temperature are found not to be important drivers of CH4 – because the seasonal temperature gradients may not have been captured – and that generally many environmental variables show weak/no relationship to methane. While the authors are correct to point out that a wide variety of factors influence fluxes, statistical power may have been low enough to limit the outcome of those analyses. Furthermore, what is measured is net flux, and concurrent production, oxidation and transport processes regulate methane flux, making interpretation more difficult.

Major Comments Figure 8. I am concerned about this plot. The relationship appears to be driven by the low CH4 exchange velocities for TMW-S (dark blue dots) however, looking at Appendix Fig B1, TMW-N has very high and variable exchange velocities which, if they were plotted, might undermine the reported relationship. If you remove the outliers from TMW-N and maintain TMW-N, do you then retain the relationship? How would this affect the results?

The conclusions are currently just a summary of the results that have already been reported. I think here there should be a greater attempt to zoom back out and generalize from the results or return to the global change context of the work.

Minor Comments Do you have concurrent CO2 observations? It appears you don't, but if you did, evaluating the CH4:CO2 ratio can provide insight into whether CH4 emissions are being limited by overall carbon flow (i.e., low CO2 respiration overall) or

competing respiration processes (i.e., low CH4 in spite of high CO2).

Is the methanogen/methanotroph ratio calculated from absolute abundance or relative? In either case, is variability in just one or the other driving the ratio variability? Is it primarily shifts in importance of methanotrophs or methanogens? If so, can this permit a more specific interpretation, e.g., variation in methanotrophy explains variation in net flux.

I suggest authors could make the zone names more specific/obvious as it is hard to recall which the acronyms refer to. Perhaps: OW = Water, FSL = Mat or Sphagnum, TMW = Tamarack, MES = Shrubs, Lagg is OK. Or Zone 1,2,3,4,5 (corresponding to concentric rings). I think this more closely ties to the central objective of the study which was to evaluate spatial heterogeneity.

Transpose table 1. Columns should be variables, rows should be entries.

Figure 3. Try grouping by wetland zone rather than month, That way you can show the full timeseries in one block, easily compare among blocks and easily see the single-block dynamics.

Line-by-Line Comments Page 13, Line 26: Check units (g m-3)? I think it should be Mg m 3. Page 15, Line 16: Mean day-time air temp? Page 15, Line 18: These range from negative to positive. Page 19, Line 22: fluxes Page 26, Line 27: can you comment on how much we can interpret from Genus level differences?

---

## Author Comment (AC1) · 12 Jul 2019

**Reviewer 1**

We thank the reviewer for the insightful comments. We have addressed the comments one by one in the following section. *Reviewer comments are written in red and italics.* Our responses are written in blue.

*I am intrigued by the approach to estimate the 'methane exchange velocity'… Methane exchange velocity: The authors do not give any information about the assumptions that go in equation*

- We have improved the explanation of the derivation of methane exchange velocity. We have also changed our term to "methane transfer velocity", more commonly reported in the literature. To hold the strict definition of the concept of methane transfer velocity, we have eliminated the assumption of negligible atmospheric methane concentrations, and included the equilibrium concentration of methane in the pore-water according to Henry' Law (See Eq 3). Since the equilibrium concentrations are virtually constant, the relationships presented previously are maintained and the values for methane transfer velocity adjusted slightly.
- We have included this change in the main text:

  By combining pore-water concentration at the surface with the associated fluxes, estimations of methane transfer velocity were obtained as in previous studies in forested ponds and lakes (Holgerson et al., 2017; Schilder et al., 2016; Wanninkhof, 2014). Through this approach, the flux at the water-air interface can be calculated using the bulk formulation:

  $$FCH_4 = k\,(Cw - Ceq) \qquad\qquad \text{Eq. (1)}$$

  Where $FCH_4$ is the diffusive $CH_4$ flux (mol m$^{-2}$ s$^{-1}$), k is the $CH_4$ transfer velocity (m s$^{-1}$), Cw is the concentration of methane in the porewater at the surface (mol m$^3$), and Ceq is the concentration of CH4 in equilibrium with the atmosphere (mol m$^3$). Ceq can is calculated by multiplying the mixing ratio of $CH_4$ in the atmosphere (s) by the atmospheric pressure (P, in MPa) and by Henry's Law coefficient of equilibrium for $CH_4$ ($K_H$) of 0.067 m3 MPa mol$^{-1}$ as in eq. 2:

  $$Ceq = s\,P / K_H \qquad\qquad \text{Eq. (2)}$$

  Ceq was calculated first with a constant mixing ratio (2 ppm) and second with the value of the average of the initial concentrations of the chamber measurements associated with each flux calculation. These two methods produced nearly identical results in Ceq given the much higher values of Cw. The constant mixing ratio was chosen for the rest of the analyses given the uncertainty associated with the initial concentration from the chambers. In the case of our peat bog, Cw can be calculated by multiplying pore-water concentration ($[CH_4]$) by peat porosity, $\Phi$ (see ancillary measurements below):

  $$Cw = [CH_4]\Phi \qquad\qquad \text{Eq. (3)}$$

Where [CH4] was calculated in the top stratigraphic layer of the peat (ca. 10 cm). Finally, methane transfer velocity can be calculated as:

$$k= (FCH4)/( Cw -Ceq ) \qquad\qquad Eq. (4)$$

*It seems that a) ebullition and plant-mediated transport have to be excluded and b) the peat structure and water/air content has to be the same for all sites (i.e. diffusivity is identical as well). Thus, by default, the only remaining factor to explain fluxes is the net methane production (i.e. microbial processes). And that is indeed, what the authors find. Only after reading the whole manuscript, it becomes clear that assumption a) is fulfilled (although the high fluxes in summer 2018 are unexplained).*

- We have now included text in the abstract to make this point clearer from the beginning. The high fluxes in summer 2018 remain unexplained since unfortunately microbial data was not available for the hotspots in the Tamarack north transect. This is the text included in the abstract:
    - Ebullition and plant-mediated transport were not important sources of $CH_4$, and the peat structure and porosity were similar across the different zones of the bog. We thus conclude that differences in $CH_4$ transfer velocities, and thus fluxes, are driven by the ratio of the relative abundance of methanogens to methanotrophs close to the bog surface.

*Microbial populations and activity: The author correctly state, that their analysis only indicates the presence of microbes, not their activity (i.e. gene expression, as was done in the Lee 2014 paper, which is cited. here). However, this makes the interpretation of Fig. 8 more difficult. I would like to point out FSL-S: Fig. 7 shows that at FSL-S, very close to the top soil, methanotrophs dominate. But for the relation with 'methane exchange velocity', only top soil ratio of methanogens/methanotrophs is used (Fig. 8) – where FSL with both high ratio of methanogens/methanotrophs and high (but variable) methane exchange velocity is clearly needed for the correlation. Given these assumptions, I wonder whether the monthly porewater concentration profiles (Fig. 6 only shows the overall mean profiles) contain more information about production, oxidation and diffusive transport (i.e. the shape of the profile). If so, this can be used as further support of microbial activity as most important driver.*

- Thank you for this interesting observation. You are right that the high abundance of methanotrophs in the top profile of the FSL-S location can be confusing. Some points to clarify about this data: The first section contains the first stratigraphic layer of the core going from ~0-6 cm, while the following section encompasses a core section from ~7-16 cm. We focus on the top section because, first, this is the section where the atmospheric exchange occurs. Secondly, this section should be the most active one for both methanogens and methanotrophs (Angle, 2017) since it includes the more aerobic acrotelm as well as less well-humified peat (greater labile C availability). Both are likely to favor greater microbial abundance. The distinction between the two sections was one based on peat stratigraphy so these two layers should be distinct in many respects. We

hypothesize that the high abundance of methanotrophs may be correlated to higher root density transporting more oxygen to this section but we did not test this hypothesis. We previously calculated relationships between microbial activity and the porewater concentrations for the whole peat profile but did not find the same patterns as just considering the top profile, likely because methane consumption mainly occurs in the upper layers. We have begun to interpret the growth of the concentration profile with time to say something about production and consumption zones within the profile, but such analysis was not within the scope of this study.

- We have included this clarification in the methods section for calculation of $CH_4$ exchange velocity

*Minor comments*

*Considering the high fluxes in summer 2018, I wonder whether the starting point may already have been high (i.e. an ebullition event early on)? Could be helpful to include the graph in the Appendix*

- We have clarified in the manuscript that this is not a part of an ebullition event and have included in the appendix the raw data from the chamber to show the steady increase in concentration:

[Figure]

Fig S3. Chamber measurement during the September hotspot in the Tam-N location. Note the steady increase in concentration that indicates that ebullition was not the reason for the high magnitude of the flux at this location.

*I understand why and how you do the up-scaling of chamber fluxes. However, there really is no way of evaluating that number and given the temporal variability there is the possibility that the large integrated flux is due to that (but temperature as discussed is possible as well)*

- We understand the limitations of this scaling approach. We provide cautious interpretation regarding how this estimate can be used to study other peat bogs. We have, however, decided to keep this estimate to provide an alternative approach to evaluate the heterogeneity in peat bogs through bottom-up measurements.

*Page 17: Is there an explanation for the result, that the instantaneous water table does not have a significant effect, but the one a month earlier has?*

- The average water level data throughout 30 days prior to the flux measurement had a significant effect in $CH_4$ fluxes. This was an interesting result and the hypothesis behind it is that the methanogens are responding to average conditions in previous weeks. In particular, we hypothesize that it takes several weeks for methanogens to acclimate to new water levels after the water level has been raised. Therefore, they do not respond instantaneously to changes in water level. From an ecological perspective, it is known that the relative abundance of organisms integrates variation in abiotic drivers over a pre-measurement time window. The length of that window will be a function of the life history and longevity of the organism. Therefore, community composition lags behind that environmental change.
- We have clarified the phrasing in page 17 and included this analysis in the discussion of water level dynamics in section 4.3

*Given the importance of the methane exchange velocity, I would move the figure from the appendix up to the main text and also discuss its error (from the figure it looks like that only for FSL and TMW the estimate is significant?).*

- We have moved the graph to the main text and now focus exclusively on those measurements of methane exchange velocity that are specific to the analyses of microbiological data rather than including measurements from other locations/months that were not used in the microbiological analysis.
- We have noted that the error in this bar plot is not being transferred to the relationship in Fig 8, since we have decided to plot the individual points rather than the average presented in this figure.

---

## Author Comment (AC2) · 15 Jul 2019

**Reviewer 2**

We thank the reviewer for the insightful comments. We have addressed the comments one by one in the following section. *Reviewer comments are written in red and italics.* Our responses are written in blue.

*The weakness of the study is that the temporal coverage and frequency of flux observations is relatively scarce despite the well-known high variability associated with methane flux. The latter means both that drivers such as temperature are found not to be important drivers of CH4 – because the seasonal temperature gradients may not have been captured – and that generally many environmental variables show weak/no relationship to methane*

- The primary objective of our study was to understand controls on spatial (intra-bog) variation in CH4 fluxes and concentrations. We agree that the lack of temporal variation is a limitation but this does not limit our ability to address biotic and abiotic drivers of differences in methane within our site. The lack of relationships between temperature and $CH_4$ flux is likely a function of our focus on variation within the late spring and summer. That approach is justifiable as spatial differences are likely to be most apparent during periods of maximum microbiological activity. It would certainly be interesting to assess intra- and inter-annual temporal variation but that would require further study and was not our aim. The spatial representation of different land covers we have investigated provides new insights into how heterogeneous $CH_4$ fluxes can be.

*While the authors are correct to point out that a wide variety of factors influence fluxes, statistical power may have been low enough to limit the outcome of those analyses. Furthermore, what is measured is net flux, and concurrent production, oxidation and transport processes regulate methane flux, making interpretation more difficult*

- Given the lack of temporal representativeness, we are cautious with our interpretation of how environmental factors affect methane fluxes and instead have focused on understanding the relationships with the microbiology and the pore-water concentrations. Through his analysis we can attempt to understand a little more about the processes of methane production and oxidation within the profile.

*Figure 8. I am concerned about this plot. The relationship appears to be driven by the low CH4 exchange velocities for TMW-S (dark blue dots) however, looking at Appendix Fig B1, TMW-N has very high and variable exchange velocities which, if they were plotted, might undermine the reported relationship. If you remove the outliers from TMW-N and maintain TMW-N, do you then retain the relationship? How would this affect the results?*

- The $CH_4$ exchange velocity in the Tamarack zone was very different between transect south and transect north. Unfortunately, microbial samples were taken only at the Tamarack south transect, so the high $CH_4$ exchange velocity of the north transect does not have microbial data to compare with. If our relationships holds true for this data point we would expect to see a very high ratio of methanogens vs methanotrophs at this northern section.

- We have now clarified this by including some of the information above in the caption of the methane transfer velocity figure

*The conclusions are currently just a summary of the results that have already been reported. I think here there should be a greater attempt to zoom back out and generalize from the results or return to the global change context of the work.*

- We have now added a paragraph in the conclusion that attempts to generalize patterns in methane release from the findings of the current studies:
    - Why would two locations with similar near-surface $CH_4$ concentrations have different fluxes if they also have similar diffusivities and negligible ebullition and plant transport? Our results show the answer is that they have different transfer velocities for $CH_4$. Transfer velocities are normally a function of wind speed, but beneath the shrub and tree canopy of peat bogs wind speeds are very low so something else is affecting this transfer velocity. The upper layer of the bog's peat mass is a dynamic region with both methanotrophs and methanogens living within the oxic layer (Angle et al., 2017). Within this layer higher abundance of methanogens drive higher transfer velocities if the concentration of $CH_4$ is assumed to be at quasi-steady state. At the same time, however, methanotrophs consume much of the methane produced. Therefore, methanogen abundance, when normalized by methanotroph abundance, can explain $CH_4$ transfer velocity differences in a peat bog where diffusive transport from porewater in saturated layers is dominant. We conclude that microbial communities, and their control by variation in water table depth, are the key drivers of variability in $CH_4$ fluxes across multiple hydro-biological zones in kettle-hole peat bogs. Future research should examine whether such patterns can be confirmed in other ecosystems where plant-mediated transport of $CH_4$ is low.

*Minor Comments*

*Do you have concurrent CO2 observations? It appears you don't, but if you did, evaluating the CH4:CO2 ratio can provide insight into whether CH4 emissions are being limited by overall carbon flow (i.e., low CO2 respiration overall) or competing respiration processes (i.e., low CH4 in spite of high CO2)*

- We do have concurrent $CO_2$ observations and have had a look at them (see fig below). To your original question, I think the lower ratios in the restored and lagg zones indicate that there is an overall low carbon availability in these zones, which is in accordance with the expectations of the level of organic matter oxidation in these zones. I think it is interesting that the ratios are all very similar at the top of the profiles but then there is a differentiation of undisturbed bog versus restored bog (RES) with depth. This probably is an indication of how the high carbon content of the bog favors methanogens at the deepest sections. We have added this information to the supplementary information.

[Figure]

*Is the methanogen/methanotroph ratio calculated from absolute abundance or relative? In either case, is variability in just one or the other driving the ratio variability? Is it primarily shifts in importance of methanotrophs or methanogens? If so, can this permit a more specific interpretation, e.g., variation in methanotrophy explains variation in net flux.*

- The ratios are calculated from the relative abundances as displayed in Fig 7. Since the relative abundance of methanotrophs is overall lower than the relative abundance of methanogens, one could expect that the variability in $CH_4$ exchange velocity is mostly driven by methanotrophs relative abundance, but that is not the case. Here are some plots

showing how the differences are not quite explained by a methanotrophs alone or methanogens alone.

[Figure]

*I suggest authors could make the zone names more specific/obvious as it is hard to recall which the acronyms refer to. Perhaps: OW = Water, FSL = Mat or Sphagnum, TMW = Tamarack, MES = Shrubs, Lagg is OK. Or Zone 1,2,3,4,5 (corresponding to concentric rings). I think this more closely ties to the central objective of the study which was to evaluate spatial heterogeneity.*

- We have adopted the first suggested change. We agree that the new labels makes the units easier to recall. Thank you for the suggestion.

*Transpose table 1. Columns should be variables, rows should be entries.*

- Done

*Figure 3. Try grouping by wetland zone rather than month, That way you can show the full timeseries in one block, easily compare among blocks and easily see the singleblock dynamics.*

- We have done it. Thank you for your suggestion.

[Figure]

*Line-by-Line Comments Page 13, Line 26: Check units (g m-3)? I think it should be Mg m 3.*

There was a typo. It was Mg m3, thank you.

*Page 15, Line 16: Mean day-time air temp?*

It is full day temperature as taken by the stations mentioned in the methods. We have clarified this in the text.

*Page 15, Line 18: These range from negative to positive.*

Fixed

*Page 19, Line 22: fluxes*

Fixed

*Page 26, Line 27: can you comment on how much we can interpret from Genus level differences?*

- Thank you, we believe the reviewer is asking specifically about the difference among the acetoclastic genera and their ecology. We have added a final clause to this paragraph commenting on the differing present of *Methanosarcina*; new text is bolded here for clarity: "**When acetoclasts were present, *Methanosaeta* dominated their community, consistent with observations of *Methanosaeta* in nutrient-poor acidic sites (Godin et al. 2012). However, in the inundated zones, *Methanosarcina* was also present. This is actually the opposite pattern we would have expected based purely on likely oxygen concentrations, as *Methanosaeta* typically dominates anaerobic environments while *Methanosarcina* can produce methane under partially oxic conditions (Angle et al 2011). We therefore interpret *Methanosaeta*'s presence in FSL-S and TMW-S to arise from its greater metabolic versatility – in addition to acetate, it can also use CO2 or methylated compounds (Liu and Whitman, 2008) – and thus that these sites may have distinct substrate profiles.**"

- It is also possible that the reviewer is asking what genus-level differences in general imply vis-à-vis e.g. function, or what the differences in these particular genera are, and so address both here. For the former: metabolism follows relatedness to varying degrees for different types of metabolism and microbes. For example, antibiotic resistance is a well-known example of a trait (sometimes metabolic) that can move dynamically among many microbial lineages, such that two closely related strains can have quite different susceptibilities to antibiotics. Other traits, such as methanogenesis, are more narrowly phylogenetically distributed and their specific methanogenic metabolisms tend to be inherited vertically (i.e. from parent to progeny cell, not acquired from other unrelated cells in the environment) and reliably. This heterogeneous relationship between metabolism and phylogeny has been reviewed for example in Martiny et al, 2015, "Microbiomes in light of traits: a phylogenetic perspective", *Science*. In the case of this research, we are examining methanogens (as noted above, for which metabolic traits do follow phylogeny in a fairly consistent way), and in addition these lineages have a large number of cultured representatives whose physiology is well-studied, and have have been ecologically characterized over decades in a variety of habitats. So, while much of microbiome science is still charting unknown waters, in the case of these dominant acetoclastic and hydrogenotrophic methanogenic genera, much is known.

References

Angel R, Matthies D, Conrad R. 2011. Activation of methanogenesis in arid biological soil crusts despite the presence of oxygen. PLoS One **6**:e20453. doi:10.1371/journal.pone.0020453.

Liu Y, Whitman WB. 2008. Metabolic, phylogenetic, and ecological diversity of the methanogenic archaea. Ann N Y Acad Sci 1125:171–189.doi:10.1196/annals.1419.019